# Fusaric acid mediates the assembly of disease-suppressive rhizosphere microbiota via induced shifts in plant root exudates

Xue Jin[1], Huiting Jia[1], Lingyi Ran[1], Fengzhi Wu[1], Junjie Liu[2], Klaus Schlaeppi [3], Francisco Dini-Andreote [4,5], Zhong Wei [6] ✉ & Xingang Zhou [1] ✉

The plant health status is determined by the interplay of plant-pathogen-microbiota in the rhizosphere. Here, we investigate this tripartite system focusing on the pathogen *Fusarium oxysporum* f. sp. *lycopersici* (FOL) and tomato plants as a model system. First, we explore differences in tomato genotype resistance to FOL potentially associated with the differential recruitment of plant-protective rhizosphere taxa. Second, we show the production of fusaric acid by FOL to trigger systemic changes in the rhizosphere microbiota. Specifically, we show this molecule to have opposite effects on the recruitment of rhizosphere disease-suppressive taxa in the resistant and susceptible genotypes. Last, we elucidate that FOL and fusaric acid induce changes in the tomato root exudation with direct effects on the recruitment of specific disease-suppressive taxa. Our study unravels a mechanism mediating plant rhizosphere assembly and disease suppression by integrating plant physiological responses to microbial-mediated mechanisms in the rhizosphere.

Plants dynamically interact with diverse microorganisms in the rhizosphere, ranging from pathogenic to commensal and mutualistic[1–4]. Soil-borne pathogens constitute major threats to plant health and survival and are responsible for significant economic losses in agriculture[5,6]. In particular, Fusarium wilt disease caused by the plant pathogen *Fusarium oxysporum*, is an important yield-limiting factor in diverse crop systems[4,7–10]. This pathogen infection starts with the initial asymptomatic penetration into the host roots, followed by the colonization of the xylem vessels, and further movement to aboveground plant tissues, which results in a gradual plant wilting[11]. On the plant-defense side, apart from physiological responses, the plant rhizosphere harbors diverse beneficial microbes, with specific taxa being

able to promote plant tolerance. This can occur via diverse mechanisms, such as antibiosis, ecological competition for resources and nutrients, and induced systemic resistance[12–14]. In this scenario, the plant health status is then determined by the interplay of the host plant metabolism, pathogen attacks, and plant-beneficial microbes in the rhizosphere[3,5,15,16].

Recent studies have demonstrated that plants can actively modulate the rhizosphere microbiota to their benefit, for instance, by antagonizing pathogens, reducing the damage caused by pathogens, and/or enhancing nutrient acquisition[3,13,17–19]. Besides, it has been shown that under pathogen or herbivore attacks, plants can selectively recruit specific beneficial microbes to assist in plant protection[8,20–27]. These

[1]Key Laboratory of Biology and Genetic Improvement of Horticultural Crops (Northeast Region), Ministry of Agriculture and Rural Affairs, Department of Horticulture, Northeast Agricultural University, 150030 Harbin, China. [2]Key Laboratory of Mollisols Agroecology, Northeast Institute of Geography and Agroecology, Chinese Academy of Sciences, 150081 Harbin, China. [3]Department of Environmental Sciences, University of Basel, 4056 Basel, Switzerland. [4]Department of Plant Science & Huck Institutes of the Life Sciences, The Pennsylvania State University, University Park, PA, USA. [5]The One Health Microbiome Center, Huck Institutes of the Life Sciences, The Pennsylvania State University, University Park, PA, USA. [6]Jiangsu Provincial Key Lab for Solid Organic Waste Utilization, Key Lab of Organic-based Fertilizers of China, Jiangsu Collaborative Innovation Center for Solid Organic Wastes, Educational Ministry Engineering Center of Resource-saving Fertilizers, Nanjing Agricultural University, 210095 Nanjing, China. ✉e-mail: weizhong@njau.edu.cn; xgzhou@neau.edu.cn

observations suggest that the recruitment of rhizosphere-beneficial microbes may represent a plant defense mechanism against plant pathogens, which aligns with the 'cry-for-help' hypothesis[28,29]. Plant species and genotypes are important determinants of the rhizosphere microbiota, and some species and genotypes generally have stronger associations with specific rhizosphere taxa[3,30,31]. More specifically, plant cultivars differing in resistance against certain pathogens can harbor distinct rhizosphere microbiota compositions[6,32–35]. However, it is still unknown whether resistant and susceptible cultivars differ in their abilities to recruit plant-beneficial microbes upon pathogen infection.

Plants have evolved a multilayered innate immune system of defense against pathogens[5]. During host infection, pathogens employ a variety of virulence factors, such as effector proteins, mycotoxins, and plant cell wall-degrading enzymes, to bypass the plant's physical barriers and suppress immune responses to facilitate disease development[36,37]. For instance, fusaric acid (FA), a secondary metabolite secreted by several pathogenic *Fusarium* species, exhibits phytotoxic activity, induces wilt symptoms in plants, and is therefore, a well-known virulence factor of fungal pathogens[7,38,39]. Emerging evidence indicates that plant pathogens can utilize virulence factors to alter the host rhizosphere microbiota and prime disease development[10,16,40,41]. For example, the effector proteins VdAve1 and VdAMP2 secreted by the fungal soil-borne pathogen *Verticillium dahliae* were shown to exhibit selective bactericidal activity against plant-beneficial microbes (e.g., *Sphingomonads*), thus directly impacting the plant rhizosphere microbiota[16]. In addition, it is also plausible that some soil-borne pathogens can indirectly affect the plant's ability to recruit beneficial microbes, with implications for infection and disease development[42].

Here, we hypothesized that (*i*) resistant and susceptible tomato cultivars (*Solanum lycopersicum*) differ in their ability to recruit plant-beneficial microbes upon pathogen infection, and (*ii*) microbial recruitment can be altered by pathogen virulence factors. To test these hypotheses, we used tomato plants and the pathogen *F. oxysporum* f. sp. *lycopersici* (FOL, causing Fusarium wilt disease) as a model system to (*1*) investigate the extent to which FOL infection differentially alters the rhizosphere microbiota of a susceptible and a resistant tomato cultivar; (*2*) determine the effect of the FOL virulence factor FA in altering tomato rhizosphere microbiota; and (*3*) evaluate how changes in root exudates alter the root colonization by beneficial bacteria.

## Results

### Differences in FOL resistance between susceptible and resistant cultivars
We initially tested the performance of two tomato cultivars (Z19 and D72) when challenged with FOL in natural (with the presence of soil microbiota) or sterile soil (absence of soil microbiota). Results showed that the cultivar Z19 has greater resistance to FOL than cultivar D72 (Tukey's HSD test, $P < 0.05$; Fig. 1a). The resistant cultivar had lower disease severity and pathogen density in natural soil than in sterile soil (Tukey's HSD test, $P < 0.05$; Fig. 1a and Supplementary Fig. 1), while no significant differences were observed between these two conditions in the susceptible cultivar. By calculating the difference in disease severity of plants grown in natural and sterile soils, we found the microbiota-mediated disease resistance to be greater in the resistant than in the susceptible cultivar (Welch's *t*-test, $P < 0.01$; Fig. 1b).

### The importance of the rhizosphere microbiota on FOL disease suppression
We used a split-root system to isolate the direct and indirect effects of FOL on the tomato rhizosphere microbiota (Fig. 1c and Supplementary Fig. 2a, b). For that, tomato plants were transplanted into two pots, with part of the root system grown in each pot. One part of the root system was inoculated with FOL (local pot), and the other part was left non-inoculated (systemic pot). For the control treatment, both sides of the root system were untreated. In brief, the disease symptoms

became apparent at fifteen and nine days after FOL inoculation for the resistant and susceptible cultivars, respectively (Supplementary Fig. 2c). At an early stage of the infection (*i.e.*, six days after FOL inoculation), FOL was only detected in the inoculated local pots but not the systemic pots (Supplementary Fig. 2d).

The disease-suppressiveness of the rhizosphere microbiota at this infection stage was tested using a rhizosphere transplant experiment (Supplementary Fig. 2a and Fig. 1c). We used sterile field soil amended with an inoculum of rhizosphere samples (6% w/w) from the systemic pots to avoid transfer of FOL, and grew a new generation of tomato plants, which were infected with FOL when they were 60 days old. Here, we found a higher disease severity and FOL abundance in the susceptible cultivar, when transplanted with the rhizosphere sample from previously infected susceptible plants (Tukey's HSD test, $P < 0.05$; Fig. 1d and Supplementary Fig. 3a). Conversely, the resistant cultivar had significantly lower disease severity and FOL abundance when plants were transplanted with the rhizosphere sample from previously infected resistant plants (Tukey's HSD test, $P < 0.05$). Moreover, the susceptible cultivar had lower disease severity and FOL abundance when plants were transplanted with the rhizosphere sample from previously infected resistant plants (Tukey's HSD test, $P < 0.05$; Supplementary Fig. 3b).

### FA produced by FOL differentially affects the rhizosphere microbiota of susceptible and resistant cultivars
We tested the potential effect(s) of the virulence factor FA on altering the rhizosphere microbiota of the resistant and susceptible cultivars. First, we found that FA was detected in the culture filtrate of FOL (Fig. 1e), and the rhizosphere samples of FOL-inoculated local pots but not the systemic pots (Fig. 1f). Next, we tested the effects of FA amendment on the disease-suppressiveness of the rhizosphere microbiota using the split-root system and a rhizosphere transplant experiment (Fig. 1g). In line with the experiment treated with FOL, similar patterns of disease severity and pathogen density were found in the experiment treated with FA. In brief, we found opposite performances of the susceptible and resistant tomato cultivars in Fusarium wilt disease severity and FOL abundance in the rhizosphere (Tukey's HSD test, $P < 0.05$; Fig. 1h, Supplementary Fig. 3c).

### FOL infection alters the composition of the tomato rhizosphere microbiota
To test whether FOL infection systemically alters the tomato rhizosphere microbiota, we used bacterial 16S rRNA gene amplicon sequencing to profile the rhizosphere bacterial community in the systemic pot from the experiment inoculated with FOL (Fig. 1c). The results revealed tomato cultivar, FOL infection, and their interaction to significantly explain variation in bacterial community β-diversity (two-way permutational multivariate analysis of variance PERMANOVA, cultivar $R^2 = 0.220$, $P < 0.001$; FOL infection $R^2 = 0.118$, $P < 0.01$; interaction effect $R^2 = 0.126$, $P < 0.01$; Fig. 2a). At the level of operational taxonomic units (OTUs), FOL infection resulted in higher relative abundances (*i.e.*, systemic pot > non-inoculated control) of 36 OTUs (29 OTUs in the susceptible cultivar and 17 OTUs in the resistant cultivar), while 30 OTUs had lower relative abundances (*i.e.*, systemic pot <non-inoculated controls) (24 OTUs in the susceptible cultivar and 7 OTUs in the resistant cultivar) (Wald test, Benjamini-Hochberg adjusted $P < 0.01$; Supplementary Fig. 4). For example, FOL infection 'enriched' three *Streptomyces* sp. OTUs in both cultivars and two *Streptomyces* sp. OTUs and one *Arthrobacter* sp. OTU in the susceptible cultivar (Fig. 2b). On the contrary, FOL infection 'depleted' one *Lysobacter* sp. OTU, three *Sphingomonas* sp. OTUs, and two *Flavobacterium* sp. OTUs in the susceptible cultivar. By comparing the two cultivars, we found a few OTUs with contrasting relative abundance changes in response to FOL infection: three OTUs (*Lysobacter* sp. OTU3848, *Sphingomonas* sp. OTU1379, and *Sphingobium* sp. OTU4099) were decreased in the susceptible

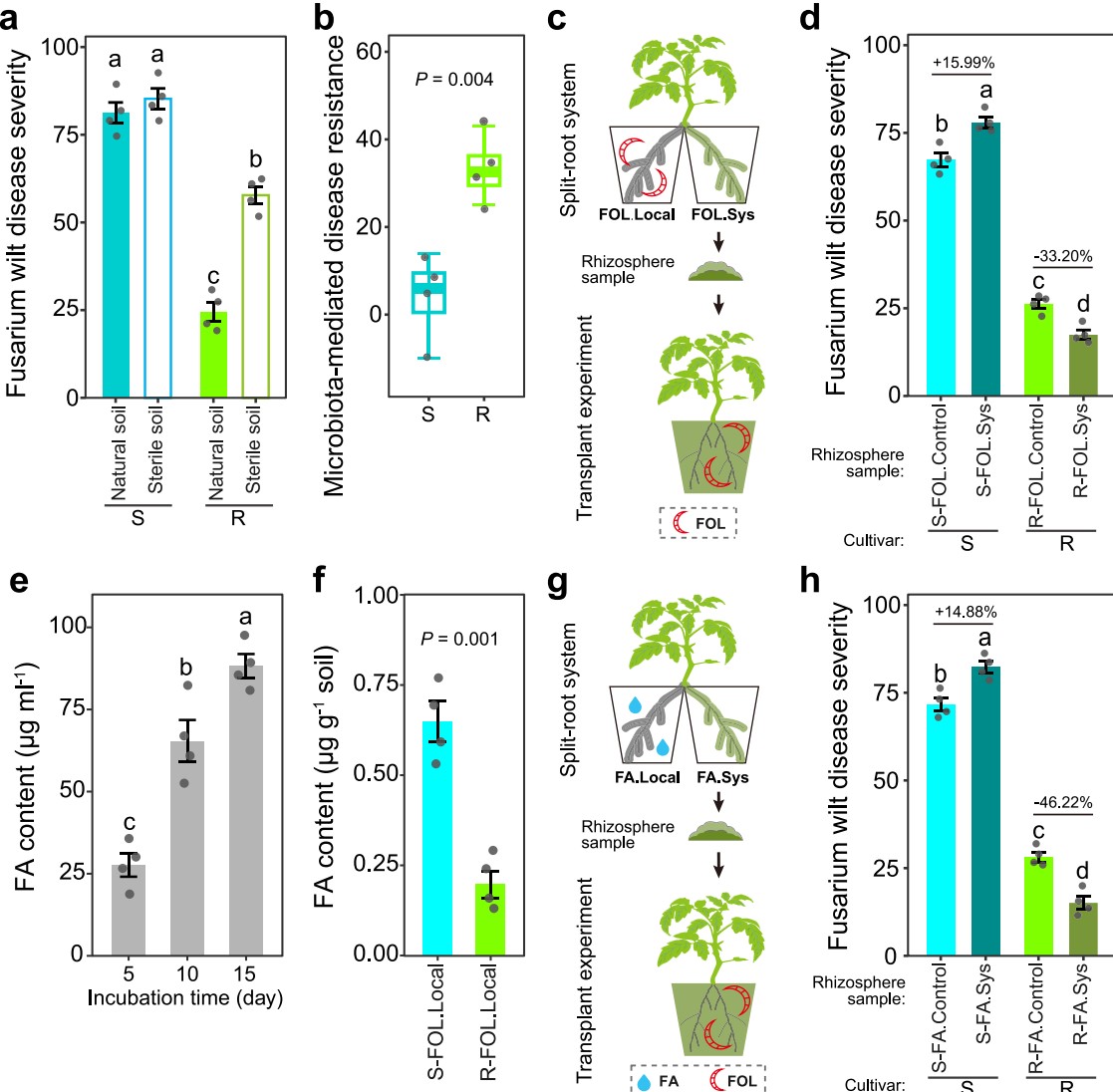

**Fig. 1 | FOL and FA-induced tomato rhizosphere microbiota recruitment and Fusarium wilt disease resistance. a** Fusarium wilt disease severity of tomato plants grown in natural and sterile soils, respectively. S, susceptible cultivar D72; R, resistant cultivar Z19. **b** Contribution of microbiota-mediated disease resistance. **c** Schematic representation of the split-root system and rhizosphere transplant experiments to test the effect(s) of FOL on the tomato rhizosphere microbiota. **d** Fusarium wilt disease severity in the rhizosphere transplant experiment used to test the effect(s) of FOL on the tomato rhizosphere disease suppressiveness. Rhizosphere samples were collected from the systemic pot of FOL-inoculated plants (FOL.Sys) and from the non-inoculated control (FOL.Control). **e** FA content in the culture filtrate of FOL grown in potato dextrose broth. **f** FA content in the tomato rhizosphere in the split-root system used to test the effects of FOL on the rhizosphere microbiota. **g** Schematic representation of the split-root system

and the rhizosphere transplant experiment used to test the effect(s) of FA on the rhizosphere disease suppressiveness. **h** Fusarium wilt disease severity in the rhizosphere transplant experiment used to test the effect(s) of FA on the rhizosphere disease suppressiveness. Rhizosphere samples were collected from the systemic pot of FA-treated plants (FA.Sys) and from the untreated control (FA.Control). FOL, *F. oxysporum* f. sp. *lycopersici*; FA, fusaric acid; Sys, systemic. In box plots, the center line represents the median, box edges delimit lower and upper quartiles and whiskers show the highest and lowest values. For (**a**, **b**, **d**–**f**, **h**), data are shown as mean ± SEM (*n* = 4). Different letters represent significant differences between treatments (Tukey's HSD test, *P* < 0.05). ns, non-significant. *P* values were determined through two-sided Welch's *t* tests. The tomato plant image was created with BioRender.com. Source data are provided as a Source Data file.

---

cultivar but increased in the resistant cultivar, while two OTUs (OTU797 and OTU2383, family Burkholderiaceae) were increased in the susceptible cultivar but decreased in the resistant cultivar (Wald test, *P* < 0.01; Fig. 2b and Supplementary Fig. 4). Further analysis at the genus level confirmed these shifts of taxa belonging to the genera *Sphingomonas*, *Sphingobium*, and *Lysobacter* in the susceptible cultivar vs. the resistant cultivar (Wald test, *P* < 0.01; Supplementary Fig. 5a).

### Fusaric acid alters the composition of the tomato rhizosphere microbiota

To evaluate whether FA alters the tomato rhizosphere bacterial community, we profiled the systemic pot from the experiment amended

with FA (Fig. 1g). The results revealed tomato cultivar, FA amendment, and their interaction to significantly explain variation in bacterial community β-diversity (two-way PERMANOVA, cultivar $R^2$ = 0.204, *P* < 0.001; FA $R^2$ = 0.138, *P* < 0.001; interaction effect $R^2$ = 0.114, *P* < 0.01; Fig. 2c). We designated OTUs in this experiment (amendment with FA) as FA.OTUs. Particularly, *Lysobacter*, *Sphingobium*, *Sphingomonas* spp., and OTUs belonging to these genera (*i.e.*, FA.OTU2652, FA.OTU2838, and FA.OTU993, respectively) decreased in relative abundances in the susceptible cultivar, and increased in the resistant cultivar (Wald test, *P* < 0.01; Fig. 2d). When comparing the bacterial genera that responded to FA amendment with those that responded to FOL infection, we found corroborating results (Supplementary Fig. 5a).

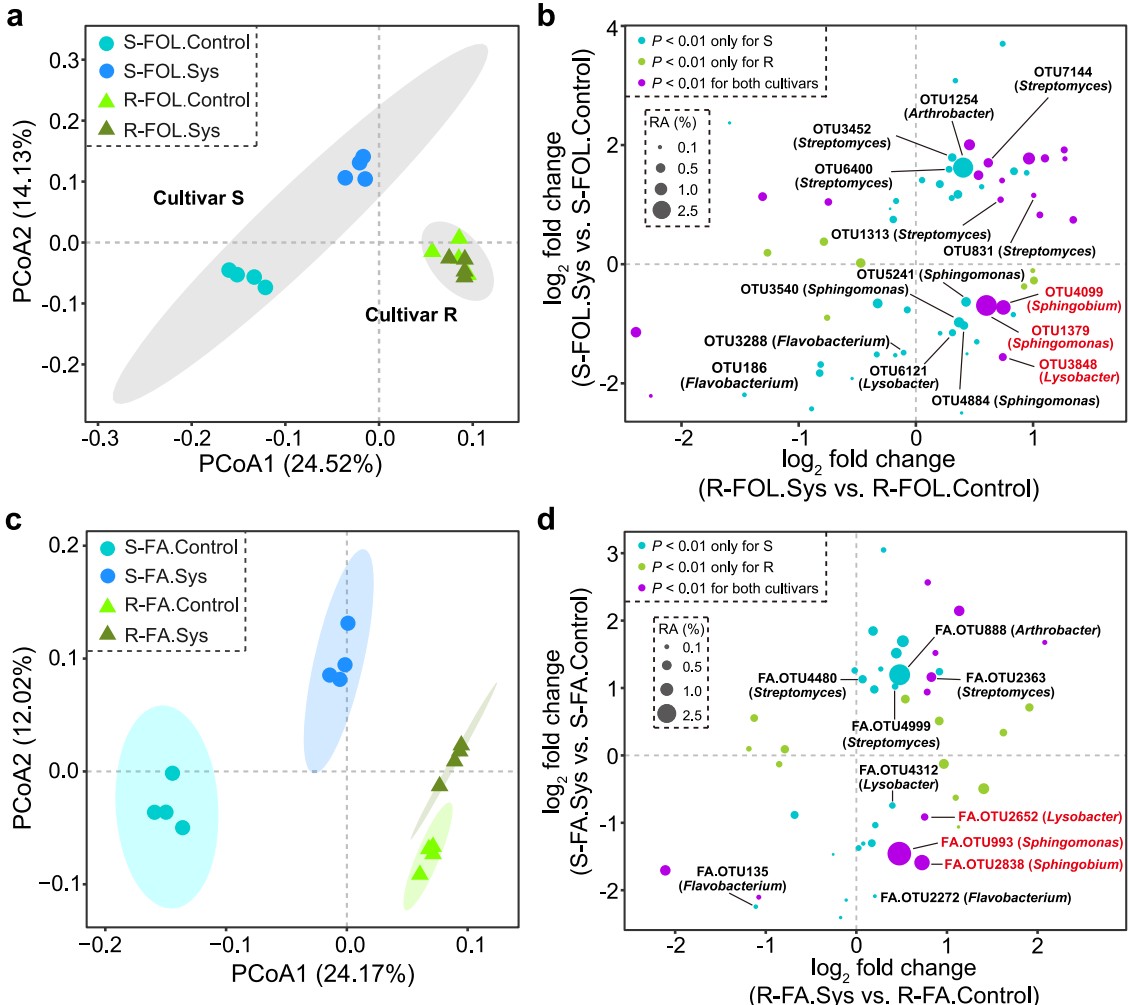

**Fig. 2 | FOL infection and FA alter the tomato rhizosphere bacterial community composition. a** Effects of FOL on the β-diversity of bacterial communities in the tomato rhizosphere of susceptible (S) and resistant (R) cultivars. **b** Comparison of bacterial OTUs altered due to FOL in the two tomato cultivars. **c** Effects of FA on the β-diversity of bacterial communities in the tomato rhizosphere of susceptible and resistant cultivars. **d** Comparison of bacterial OTUs altered due to FA in the two tomato cultivars. For (**b**) and (**d**), data are shown as the log₂-fold differences between FOL.Sys vs. FOL.Control (or FA.Sys vs. FA.Control) for each cultivar (two-sided Wald test with $P$ values Benjamini-Hochberg adjusted). The size of each circle corresponds to its mean relative abundance (RA) across all samples. The color of each circle represents whether the taxon is altered by FOL (or FA) in only one cultivar or in both cultivars. FOL, *F. oxysporum* f. sp. *lycopersici*; FA, fusaric acid; Sys, systemic. Source data are provided as a Source Data file.

For example, 9/16 and 8/14 genera that increased in FOL inoculated plants, also had similar responses due to FA amendment in the susceptible and resistant cultivars, respectively. Likewise, 14/30 and 3/8 genera with lower relative abundances in FOL-inoculated plants, also had similar responses due to FA amendment in the susceptible and resistant cultivars, respectively (Supplementary Fig. 5b). Last, co-clustering of the sequencing data revealed that FOL and FA decreased the relative abundances of the exact same OTUs belonging to *Lysobacter*, *Sphingobium*, and *Sphingomonas* spp. in the susceptible cultivar, and increased the relative abundances of these OTUs in the resistant cultivar (Supplementary Fig. 6).

**Validation of beneficial bacterial taxa acting on disease suppression**

We isolated and identified a total of 573 bacterial isolates from the tomato rhizosphere, and focused our analysis on bacterial isolates belonging to the genera *Flavobacterium* (6 isolates), *Arthrobacter* (6 isolates), *Streptomyces* (10 isolates), *Lysobacter* (5 isolates), *Sphingobium* (5 isolates), and *Sphingomonas* (7 isolates) (Fig. 3a). Most of these isolates—except for one *Flavobacterium*, one *Arthrobacter*, and two

*Streptomyces* isolates—had the ability to reduce the symptoms of Fusarium wilt disease in both cultivars (Tukey's HSD test, $P < 0.05$; Fig. 3b, c). Particularly, the tested *Sphingomonas* isolates had the greatest disease suppression across all tested isolates (Tukey's HSD test, $P < 0.05$).

We further selected one isolate of each genus that mapped most closely to their corresponding OTU sequences; *i.e.*, *Flavobacterium* sp. Fl79 (OTU3288), *Arthrobacter* sp. Ar03 (OTU1254), *Streptomyces* sp. St81 (OTU1313), *Lysobacter* sp. Ly56 (OTU3848, 99.77%), *Sphingobium* sp. Sb87 (OTU4099), and *Sphingomonas* sp. Sm12 (OTU1379) (Fig. 3a). These selected isolates were used to build a synthetic community (SynCom) and tested for its potential to promote Fusarium wilt disease suppression in a pot experiment (Fig. 3d). In addition, we performed drop-out SynComs (excluding all possible individual isolates from this SynCom) to test the contribution of each isolate to the disease suppression. The complete SynCom resulted in a decreased Fusarium wilt disease severity of 65.51% and 84.39% for the susceptible and resistant cultivars, respectively (Tukey's HSD test, $P < 0.05$; Fig. 3d). The drop-out SynComs experiments revealed that—except for isolate Fl79 in the resistant cultivar—a lower disease suppression was

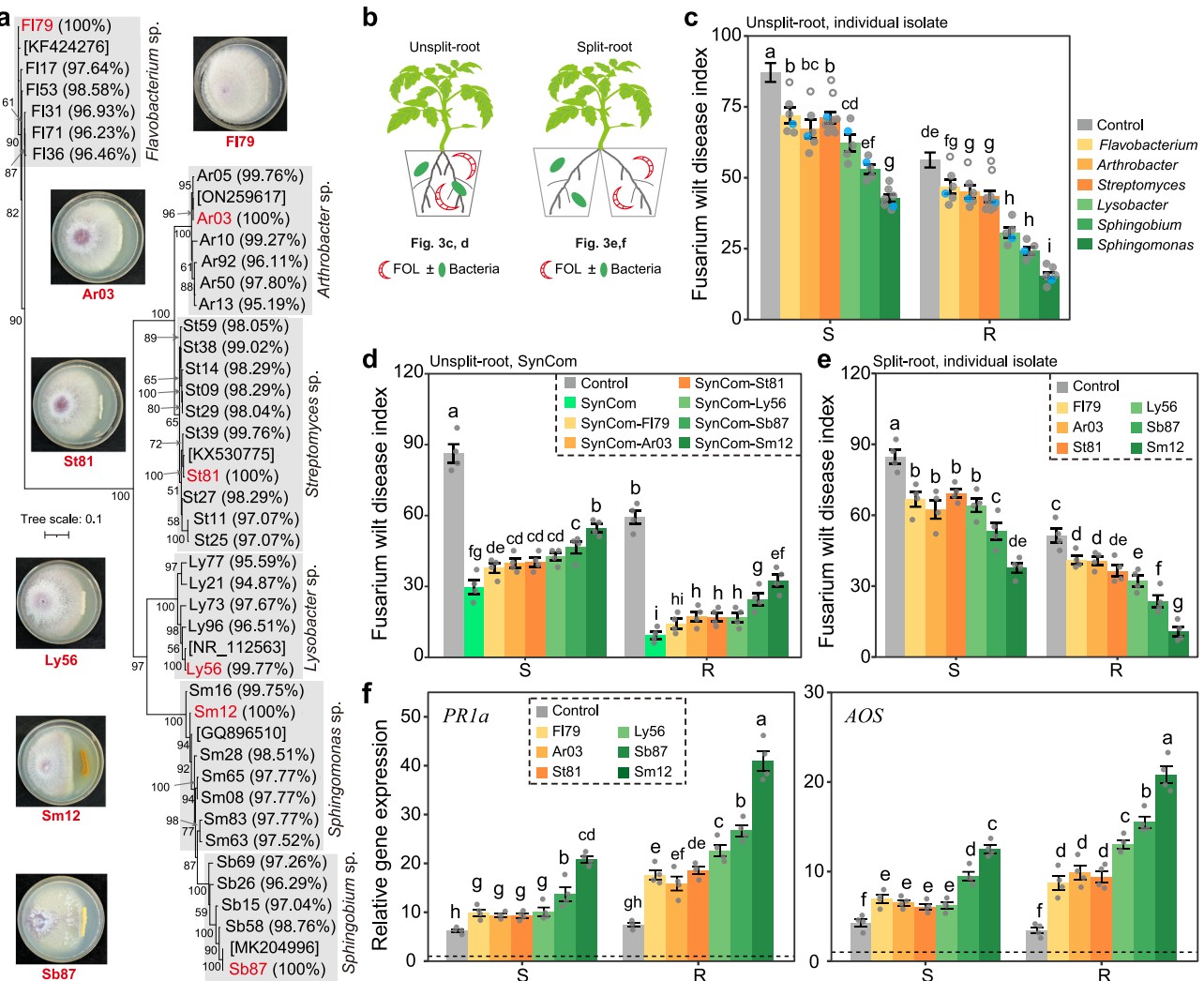

**Fig. 3 | Effects of bacterial isolates and distinct SynComs on Fusarium wilt disease suppression. a** Maximum likelihood phylogenetic tree based on the 16 S rRNA gene sequences of bacterial isolates. The sequence similarity of each *Flavobacterium*, *Arthrobacter*, *Streptomyces*, *Lysobacter*, *Sphingobium* and *Sphingomonas* spp. isolates with OTU3288, OTU1254, OTU1313, OTU3848, OTU4099 and OTU1379, respectively, is provided between parentheses. The NCBI database accession numbers of the closest phylogenetic relatives of isolates having the highest sequence similarity with these OTUs (in red color) are provided in brackets. Photos on the right panel show the in vitro antagonistic activity of selected isolates against FOL on potato dextrose agar. **b** Schematic representation of the pot experiment used to test the suppressive potential of bacterial isolates and Syn-Coms on pathogen FOL. **c** Effects of individual bacterial isolates on Fusarium wilt disease severity. S, susceptible cultivar D72; R, resistant cultivar Z19. **d** Effects of the

SynComs on Fusarium wilt disease severity. **e** Effects of bacterial-induced systemic resistance against Fusarium wilt disease. **f** Relative expression levels of defense-related genes in tomato roots. For each tomato cultivar, results are normalized to tomato *ACTIN* gene and expressed relative to those detected in non-inoculated plants before FOL infection–set as an arbitrary value of 1 (dashed line). FOL, *F. oxysporum* f. sp. *lycopersici*; SynCom, synthetic community. For (**c**), each dot represents one bacterial isolate ($n = 4$ for each isolate). Solid and open dots represent isolates with significant and non-significant disease suppressive effects (as compared with the non-inoculated control treatment). Blue dots represent the isolates Fl79, Ar03, St81, Ly56, Sb87, or Sm12. For (**d**–**f**), data are shown as mean ± SEM ($n = 4$). Different letters represent significant differences between treatments (Tukey's HSD test; $P < 0.05$). The tomato plant image was created with BioRender.com. Source data are provided as a Source Data file.

consistently observed. Besides, these experiments revealed *Sphingomonas* sp. Sm12 to have the greatest effect on disease suppression. This was consistent with an in vitro observation in which only Sm12 had an antagonistic activity against FOL when co-cultured on a potato dextrose agar Petri dish assay (Fig. 3a).

We also utilized the split-root system to test whether these bacterial isolates can promote Fusarium wilt disease suppression via induced systemic resistance (*i.e.*, one side of the pot was inoculated with a bacterium and the other with FOL; Fig. 3b). All tested isolates significantly suppressed Fusarium wilt disease severity in both cultivars, with Sm12 having the greatest suppression (Fig. 3e). The disease severity scores were consistent with the enhanced expression levels of defense-related genes ALLENE OXIDE SYNTHASE (*AOS*) and PATHOGENESIS-RELATED PROTEIN 1 (*PR1a*) in systemic roots that not

inoculated with bacteria (Tukey's HSD test, $P < 0.05$; Fig. 3f). In addition, these bacterial isolates decreased both FOL abundance and FA content in the rhizosphere from the pot inoculated with FOL (Tukey's HSD test, $P < 0.05$; Supplementary Fig. 7a). Together, we found most of these tested bacterial isolates–in particular Ly56, Sb87, and Sm12–to have a significant suppressive effect on Fusarium wilt disease via induced systemic resistance. Specifically, we found Sm12 to have the greatest disease-suppressive effect.

**The colonization of *Sphingomonas* sp. in the rhizosphere is regulated by FOL via FA secretion**
We utilized the split-root system to estimate the root colonization of these bacterial isolates in response to FOL infection (Fig. 4a). For that, one part of the root system was inoculated with each isolate or the

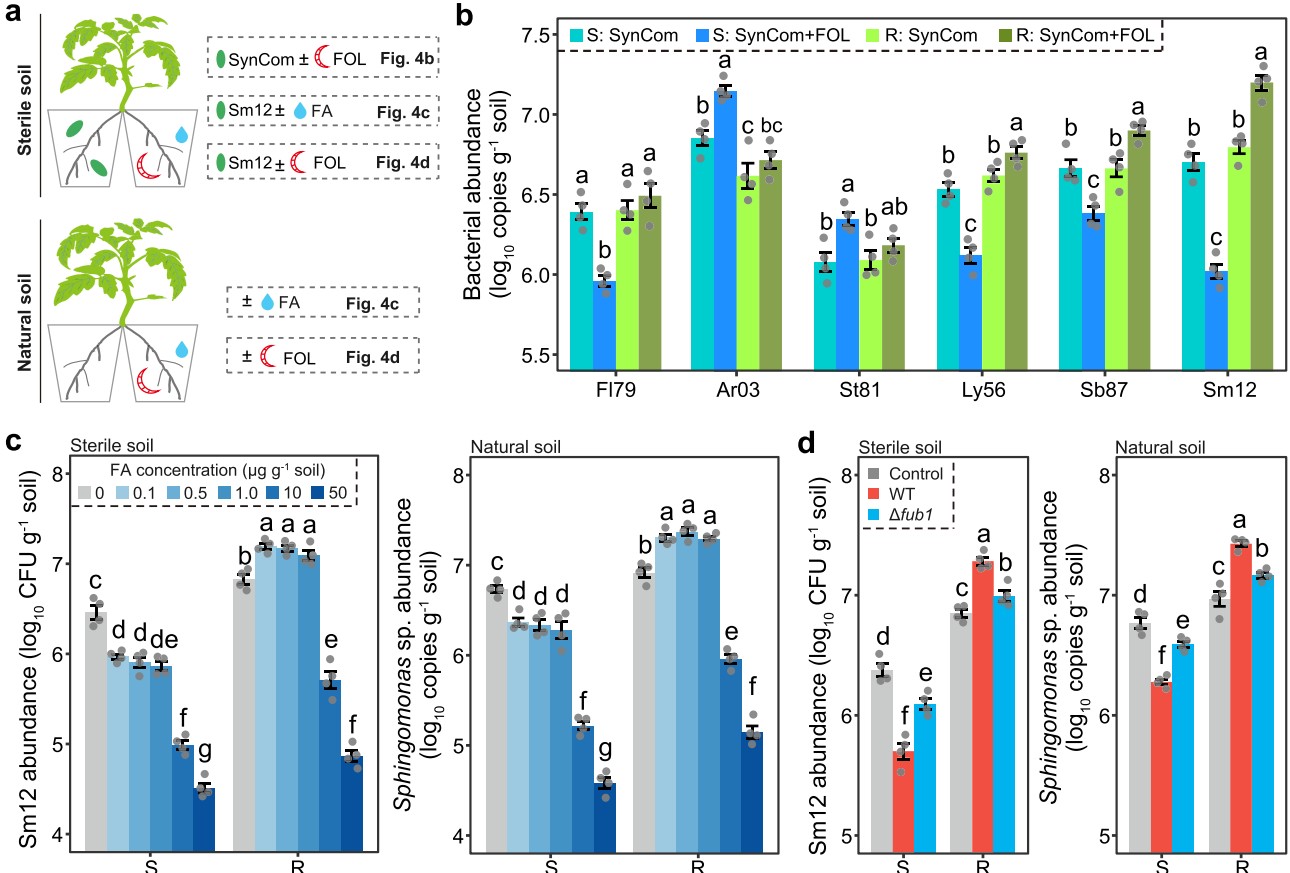

**Fig. 4 | FOL and FA modulate the root colonization by *Sphingomonas* sp.**
**a** Schematic representation of the split-root system used to test the root coloni-zation by bacterial isolates and SynComs. **b** Effects of FOL infection on the root colonization by each bacterial isolate in the SynComs. S, susceptible cultivar D72; R, resistant cultivar Z19. **c** Effect of exogenous FA amendment on the abundances of Sm12 and *Sphingomonas* sp. in the rhizosphere of tomato. **d** Effect of the FOL wild-type (WT) and Δ*fub1* mutant on the abundances of Sm12 and *Sphingomonas* sp. in the rhizosphere of tomato. FOL, *F. oxysporum* f. sp. *lycopersici*; FA, fusaric acid; SynCom, synthetic community; CFU, colony forming units. For (**b**–**d**), data are shown as mean ± SEM (*n* = 4). Different letters represent significant differences between treatments (Tukey's HSD test; *P* < 0.05). The tomato plant image was created with BioRender.com. Source data are provided as a Source Data file.

SynCom, and the other part was inoculated with FOL. Quantitative PCR revealed a consistent enrichment of Ar03 and St81 in the bacterial isolate inoculated pot. Interestingly, a cultivar-specific response of the other isolates was observed, where Ly56, Sb87, and Sm12 were reduced in the susceptible cultivar but increased in the resistant cultivar following FOL infection (Tukey's HSD test, *P* < 0.05; Supple-mentary Fig. 7b). Similar results were obtained when measuring the individual isolates after inoculation with a SynCom. In particular, we found increased abundances of Ly56, Sb87, and Sm12 in the resis-tant cultivar (Tukey's HSD, *P* < 0.05; Fig. 4b). Thus, these results corroborate the data obtained via bacterial community sequen-cing (Fig. 2d).

We further tested the potential effect of FA on root colonization by *Sphingomonas* sp. In brief, this taxon was selected as it showed the highest disease-suppressive effect. For that, we performed exogenous FA amendments in the local pot and evaluated its effect on the abun-dance of Sm12 (tomato grown in sterile soil inoculated with Sm12) and *Sphingomonas* (tomato grown in natural soil) in the rhizosphere in the systemic pot (split-root system). (Fig. 4a). We used a rifampicin-resistant strain of *Sphingomonas* sp. Sm12 as a representative bacter-ium in the assay in sterile soil and measured its abundance by plating on LB agar medium amended with 150 μg ml⁻¹ rifampicin. In the sterile soil inoculated with Sm12 system, the amendment with FA (at con-centrations detected in the tomato rhizosphere; *i.e.*, 0.1–1.0 μg g⁻¹ soil) resulted in a lower abundance of Sm12 in the rhizosphere of the sus-ceptible cultivar, but in a higher abundance in the rhizosphere of the

resistant cultivar (Tukey's HSD test, *P* < 0.05; Fig. 4c). Similarly, in the natural soil system, FA decreased the abundance of *Sphingomonas* sp. in the rhizosphere of the susceptible cultivar and increased it in the rhizosphere of the resistant cultivar. Importantly, we also noticed that high concentrations (i.e., 10 and 50 μg g⁻¹ soil) of FA negatively impact the abundance of *Sphingomonas* sp. in the rhizosphere of both culti-vars (Tukey's HSD test, *P* < 0.05). We also tested whether FA is toxic to bacteria and tomato plants. At high concentrations of 10 and 50 μg ml⁻¹, but not at concentrations ranging from 0.1 to 1.0 μg ml⁻¹, FA inhibited the in vitro growth and biofilm formation (important pro-cesses of bacterial root colonization[23]) of Sm12, and caused wilt symptoms in tomato plants (Tukey's HSD test, *P* < 0.05; Supplemen-tary Fig. 8a, b).

Last, we validate these findings by generating a FOL mutant that lacks the FA biosynthetic gene 1 (*FUB1*), encoding a REDUCING POLYKETIDE SYNTHASE[39], in the wild-type FOL. The *FUB1* deletion mutant (Δ*fub1*) lost its ability to produce FA (Supplementary Fig. 9a, b). The Δ*fub1* also caused weaker wilt disease symptoms in tomato plants compared to the wild-type (Tukey's HSD test, *P* < 0.01; Supplementary Fig. 9c). We used the split-root system to specifically test the effects of Δ*fub1* on the root colonization by Sm12 in sterile soil and by *Sphin-gomonas* sp. in natural soil (Fig. 4a). Compared with the FOL wild type, the Δ*fub1* mutant had weaker negative effects on the rhizosphere colonization by Sm12 and *Sphingomonas* sp. in the susceptible cultivar, and weaker positive effects on these taxa colonization in the resistant cultivar (Tukey's HSD test, *P* < 0.05; Fig. 4d).

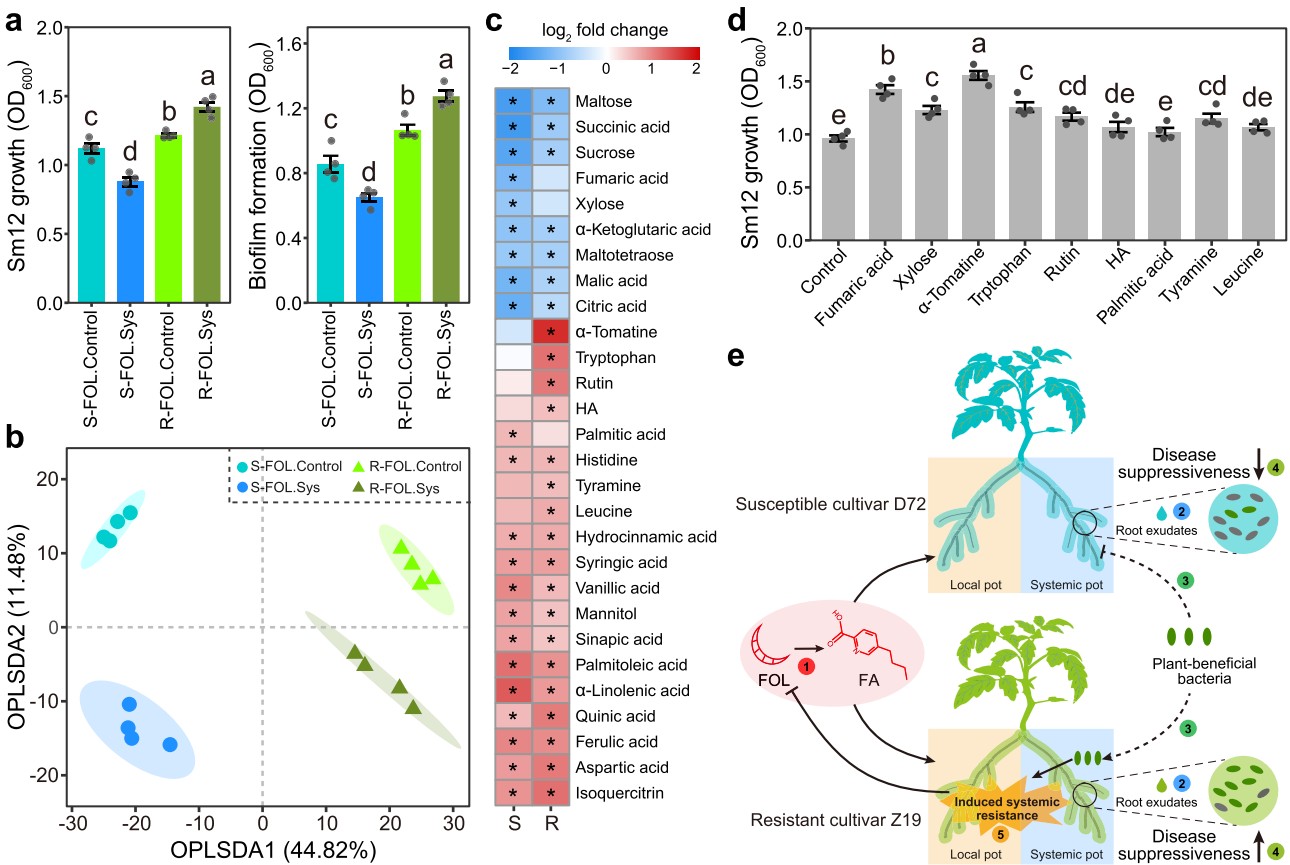

**Fig. 5 | Effect of FOL inoculation on tomato root exudates and root colonization by *Sphingomonas* sp. Sm12. a** Effects of tomato root exudates on the growth and biofilm formation of Sm12. Tomato root exudates were collected from the systemic pot of FOL-inoculated plants (FOL.Sys) and from the non-inoculated control (FOL.Control). S, susceptible cultivar D72; R, resistant cultivar Z19. **b** Orthogonal partial least squares-discriminant analysis (OPLS-DA) of metabolites in tomato root exudates. **c** Heatmap depicting the distinct metabolites in tomato root exudates between treatments. *indicates significant altered metabolites with variable importance of projection > 1, log$_2$ fold change > 1 and Benjamini-Hochberg adjusted $P < 0.01$ (two-sided Wald test). **d** Effects of specific metabolites on the growth and biofilm formation of Sm12. **e** Conceptual diagram of the mechanisms by which FOL-derived FA modulates the disease suppressive status of the tomato

rhizosphere microbiota. In brief, FOL produces FA (1), which differentially alters the tomato root exudates in susceptible and resistant cultivars (2). These differences in root exudate metabolites impact the root colonization by disease-suppressive bacterial taxa (e.g., *Sphingomonas* sp.) (3), and thus modulate the status of plant health (4). Importantly, the root colonization by these bacteria were stimulated in the resistant cultivar Z19 but inhibited in the susceptible cultivar D72. These disease-suppressive bacteria act via induced systemic resistance (5). HA, 2-Hydroxyglutaric acid; FOL, *F. oxysporum* f. sp. *lycopersici*; FA, fusaric acid; OD, optical density. For (**a**, **d**), data are shown as mean ± SEM (*n* = 4). Different letters represent significant differences between treatments (Tukey's HSD test; *P* < 0.05). The tomato plant image was created with BioRender.com. Source data are provided as a Source Data file.

## FOL and FA mediate the root colonization by *Sphingomonas* sp. via induced shifts in tomato root exudates

We tested the effect of FOL inoculation and FA amendment on shifts in tomato root exudation with a potential impact on root colonization by *Sphingomonas* sp. First, we collected tomato root exudates from the systemic pot of FOL-inoculated or FA-treated plants and from the non-inoculated control plants (Fig. 1c, g). We tested the in vitro effects of tomato root exudates on the growth and biofilm formation of Sm12. We found the root exudates of the susceptible cultivar from the systemic pot inoculated with FOL to inhibit the growth and biofilm formation of Sm12 as compared with the control (Tukey's HSD test, *P* < 0.05; Fig. 5a). On the other hand, root exudates of FOL-inoculated resistant cultivar plants had stimulatory effects on the growth and biofilm formation of Sm12. Similarly, we found that root exudates from the susceptible cultivar treated with FA had inhibitory effects, while root exudates from the resistant cultivar treated with FA had stimulatory effects on the growth and biofilm formation of Sm12 (Tukey's HSD test, *P* < 0.05; Supplementary Fig. 10a).

We further analyzed the tomato root exudates using high-pressure liquid chromatography-mass spectrometry (HPLC-MS) analysis. The orthogonal partial least squares-discriminant analysis

(OPLS-DA) model showed that FOL inoculation plants altered the overall root exudates of both cultivars (Permutation test, $R^2X = 0.470$ and $Q^2 = 0.798$, and $R^2X = 0.502$ and $Q^2 = 0.893$ for the susceptible and resistant cultivars, respectively; Fig. 5b). In total, 28 metabolites were altered by FOL in the susceptible cultivar and/or the resistant cultivar (variable importance of projection (VIP) > 1, log$_2$ fold change > 1 and Benjamini-Hochberg adjusted *P* < 0.01; Fig. 5c). In particular, FOL inoculation resulted in lower concentrations of fumaric acid and xylose, and a higher concentration of palmitic acid in the susceptible cultivar, but not in the resistant cultivar. Besides, FOL inoculation increased the concentration of α-tomatine, tryptophan, rutin, 2-hydroxyglutaric acid, tyramine, and leucine production in the root exudate of the resistant cultivar, but not in the susceptible cultivar. Analysis of tomato root exudates from the systemic pot of FA-treated plants revealed that the exogenous amendment of FA altered tomato root exudates profile (Permutation test, $R^2X = 0.457$ and $Q^2 = 0.817$, and $R^2X = 0.516$ and $Q^2 = 0.819$ for the susceptible and resistant cultivars, respectively; Supplementary Fig. 10b). Moreover, FA amendment resulted in lower concentrations of fumaric acid and xylose in the susceptible cultivar, but not in the resistant cultivar; in addition to increased concentrations of α-tomatine, rutin, tryptophan, and

tyramine in the resistant cultivar, but not in the susceptible cultivar (Supplementary Fig. 10c). Further in vitro assays found that exogenous amendments with fumaric acid, xylose, α-tomatine, tryptophan, rutin, tyramine, and leucine (at 10 μM) stimulated the growth and biofilm formation of Sm12 (Tukey's HSD test, $P < 0.05$; Fig. 5d and Supplementary Fig. 10d).

Last, root exudates from both cultivars infected or not with FOL had similar effects on FOL growth, FA production, and on the expression of the *FUB1* gene (Supplementary Fig. 11a). In addition, FA amendment did not result in changes in tomato root exudates with potential subsequent effects on FOL growth, FA production, and *FUB1* gene expression (Supplementary Fig. 11b).

## Discussion

The rhizosphere microbiota constitutes the first layer of plant defense against soil-borne pathogens[5]. Previous studies have demonstrated that pathogen infection can alter the structure of plant rhizosphere microbiota[8,20–26]. However, the mechanisms underlying such effects remain still largely unknown. Here, we show that a key molecule (FA) produced by the pathogen FOL can result in distinct impacts on the modulation of the tomato rhizosphere microbiota in susceptible and resistant cultivars. In brief, we found that—at the early stages of infection—FOL-derived FA was higher in the rhizosphere of susceptible cultivar than in that of the resistant cultivar. We further explored this to show that FA content differentially modulates the composition of root exudates across cultivars and that such effects directly affect the plant's ability to recruit beneficial microbial taxa in the rhizosphere. As such, we here report an indirect effect of FOL-derived FA in structuring a rhizosphere disease-suppressive or conducive microbiota (Fig. 5e), with trackable variations across susceptible and resistant cultivars.

Using tomato in a split-root system assay, we showed that FOL inoculation in the local pot impacts the composition and disease suppressiveness status of the rhizosphere microbiota in the systemic pot (Fig. 1). Importantly, since this systemic effect was detected in the rhizosphere—which extends beyond the host tissues into the soil—it is plausible to assume that this effect can be transferred to the next generation of plants growing in that soil[20,27]. By characterizing a collection of bacterial isolates from the tomato rhizosphere, we showed that FOL infection primarily altered the colonization of specific plant-beneficial taxa, *i.e.*, bacteria reducing FOL symptoms, in the systemic pots (Fig. 3). It has been shown that soil-borne pathogens alter the composition and functioning of plant rhizosphere microbial communities[35,43–47]. This can be achieved directly via pathogen exo-metabolite production in the rhizosphere[16,40,41] and potentially indirectly by mediating physiological changes in host plants. In this study, we unraveled an indirect pathogen-mediated effect on the rhizosphere microbiota via induced changes in tomato root exudates (Fig. 5e). Considering that the rhizosphere represents a spatially heterogeneous environment and that soil-borne pathogens only infect sections of the plant root systems[48], systemic changes in the rhizosphere microbiota induced by FOL can potentially have a significant influence on the outcome of the plant-pathogen-microbiota interaction in the soil.

We found the resistant cultivar to have a higher resistant to FOL when compared to the susceptible cultivar irrespective of soil microbiota. We believe this might be due to variation in plant innate immunity that contributes to FOL resistance across cultivars[5,11]. Moreover, the FOL-induced changes in the rhizosphere microbiota differed between the resistant and susceptible tomato cultivars. Specifically, the rhizosphere microbiota of the resistant cultivar after FOL infection was suppressive against Fusarium wilt disease, while that of the susceptible cultivar was conducive (Fig. 1). In brief, we found that FOL stimulated the resistant cultivar to recruit suppressive bacteria (such as *Sphingomonas* sp.) but impaired the recruitment of these bacteria in the susceptible cultivar. It is important to notice that even though the root colonization of *Sphingomonas* sp. was inhibited, other

taxa (e.g., *Streptomyces* and *Arthrobacter* spp.) with mild disease-suppressive potential were enriched in the susceptible cultivar upon FOL infection. Moreover, we found these recruited bacteria to promote Fusarium wilt disease suppression via induced systemic resistance. This was shown by an enhanced expression of defense-related genes, including *AOS* and *PR1a* in tomato roots. It is important to notice that our study specifically focused on rhizosphere bacterial communities. As such, other types of biotic interactions in the rhizosphere, including fungal communities and mycorrhizal associations—known to differ across plant genotypes and potentially affect disease outcomes[49,50]—were not explicitly studied in our system.

To cause disease, pathogens have to bypass the first line of plant defense, *i.e.*, the rhizosphere microbiota, for instance, by disrupting the mutualistic relationship between a plant and beneficial microbial taxa[16,51]. In line with that, we found FOL infection to inhibit the colonization of efficient disease-suppressive taxa (e.g., *Sphingomonas* sp.) in the rhizosphere of the susceptible cultivar. As such, it is plausible to assume that the capacity of FOL to cause infection and disease development is—at least partially—determined by its ability to undermine the plant's ability to recruit plant-protective taxa. Moreover, despite we show the ability of this pathogen to indirectly modulate the rhizosphere microbiota, other direct mechanisms have been previously shown. For example, the soil-borne pathogen *V. dahliae* directly inhibits plant-beneficial bacteria via secreted protein effectors with antimicrobial activities, thus promoting disease development in tomato and cotton plants (*Gossypium* spp.)[16].

Plant pathogens often employ virulence factors to counter plant immunity and cause infection[5]. However, relatively little is known about the potential of these virulence factors in impacting the rhizosphere microbiota. Here, we show that the phytotoxin FA produced by FOL acts by systemically altering the bacterial community composition and the disease-suppressive status of the tomato rhizosphere. In the resistant cultivar, the amendment with FA (at the concentration of $0.1\ \mu g\ g^{-1}$ soil) favored the recruitment of a rhizosphere-suppressive microbiota. On the other hand, this compound decreased the disease-suppressiveness of the rhizosphere microbiota in the susceptible cultivar. We found that FA concentrations in the tomato rhizosphere (ranging from 0.1 to $1.0\ \mu g\ g^{-1}$ soil) modulate root colonization by the plant-beneficial bacterium *Sphingomonas* sp. Sm12. In line with that, we and others have demonstrated that inactivation of the FA biosynthetic gene in the pathogen *F. oxysporum* reduces this pathogen's virulence[7,39], thus validating FA as a virulence factor of FOL. In addition, we show that high concentrations of FA (10 and $50\ \mu g\ g^{-1}$ soil) are toxic to tomato plants and also negatively impact bacterial growth. Thus, these results align with the previously reported dose-dependent toxicity of FA[38,52]. We also validated the direct antibacterial activity of FA at high concentrations against the disease-suppressive bacterium Sm12 (Supplementary Fig. 8a). This might also be an additional mechanism by which FA facilitates FOL infection. Therefore, besides the ability to systemically impact the tomato rhizosphere microbiota, FA can directly inhibit plant-beneficial bacteria to facilitate infection. These findings corroborate previously reported functions of the protein effectors VdAve1, VdAve1L, and VdAMP2 of *V. dahliae* that facilitate disease development[16,41]. We found that—compared with the non-inoculated control—the *Δfub1* resulted in lower abundances of Sm12 and *Sphingomonas* sp. in the susceptible cultivar, and in greater abundances in the resistant cultivar. These findings suggest that potentially other FOL virulence factors can also exert an influence on the colonization of roots by plant-beneficial bacteria. Taken together, FA can have distinct direct and indirect effects in the rhizosphere, for instance: (*1*) by acting as a molecule altering plant physiology and resulting in shifts in the rhizosphere microbiota, and/or (*2*) by negatively affecting microbial taxa, and/or (*3*) by acting as a phytotoxin that facilitates the pathogen infection and disease progression.

The susceptible and resistant tomato cultivars had distinct rhizosphere microbiota compositions, which is in line with the effect of host genotype on plant-associated microbiotas[3,31,33]. It is well known that distinct plant cultivars can differentially recruit specific bacterial taxa in the rhizosphere[3,32,53]. For example, plant-beneficial *Flavobacteriaceae* bacteria were found to be more abundant in the rhizosphere of the tomato cultivar Hawaii 7996, resistant to *Ralstonia* wilt, compared with Moneymaker, a susceptible cultivar[6]. Here, we described cultivar-specific recruitment of plant-beneficial bacteria in response to FOL infection (Fig. 2). In line with that, plant root exudates play an important role in regulating the rhizosphere microbiota assembly and functioning[13,17,54–56]. As such, alterations in plant physiology due to biotic and abiotic stresses are likely to influence the rhizosphere microbiota. Consistent with previous studies[25,57–59], we found that FOL infection altered the tomato root exudates profile. In particular, this was shown to be mediated by FOL-derived FA, which resulted in shifts in specific exudate metabolites (such as α-tomatine) found to be increased in concentration by FOL inoculation and FA amendment in the resistant but not in the susceptible cultivar. These compounds were shown to promote the growth and biofilm formation of Sm12, thus enhancing its rhizosphere competence. In addition, other compounds also had positive effects on Sm12 (such as fumaric acid). This was shown to decrease in concentration in the susceptible cultivar but not the resistant cultivar after FOL inoculation. Therefore, the differentially altered root exudate profile induced by FOL-derived FA is an additional mechanism modulating root colonization by beneficial bacteria between the susceptible and resistant cultivars (Fig. 5). Interestingly, a recent study reported that α-tomatine secreted by tomato roots was linked to the recruitment of *Sphingomonadaceae* species in the rhizosphere[56]. However, further studies (for example, based on gene knockouts of the biosynthesis of these compounds) are necessary to validate the function of these compounds in plant-microbe interactions.

Collectively, our results support that FOL infection—associated with the virulence factor FA—causes shifts in the rhizosphere microbiota, which in turn affect pathogen establishment and ultimately plant health. The rhizosphere microbiota assembly and disease suppressiveness were differentially altered by FOL in susceptible and resistant cultivars (Fig. 5e). In particular, the resistant cultivar recruits a disease-suppressive rhizosphere microbiota in response to FOL infection. In contrast, FOL impairs the ability of susceptible plants to recruit strong disease-suppressive bacterial taxa, thus facilitating pathogen infection. These FOL-induced changes in the tomato rhizosphere microbiota occur via the virulence factor FA that differentially alters the root exudate profile between cultivars, thus resulting in distinct rhizosphere microbiota compositions and disease-suppressive statuses. Our results show that the ability to recruit plant-beneficial bacteria in response to FOL infection constitutes an important first line of defense against the soil-borne pathogen. This study also highlights the importance of the plant-pathogen-microbiota tripartite interaction in determining the outcome of pathogen infections in plant-soil systems.

## Methods
### Soil, tomato plants, and FOL inoculation
The soil used in this study was collected from a cropland (top 20 cm) in Xiangyang County, Harbin, China (45°78′N, 126°94′E). This field has been cultivated with maize for the past nine years. This soil was not previously cultivated with tomato plants to prevent potential effects of high FOL densities in our experimental systems. The collected soil was sieved (<2 mm) and analyzed for physicochemical properties. In brief, this soil was characterized by a pH of 7.18 (1:2.5, w/v), electrical conductivity of 0.30 mS cm$^{-1}$ (1:2.5, w/v), 37.55 g kg$^{-1}$ of organic matter, 88.63 mg kg$^{-1}$ of inorganic nitrogen (NH$_4^+$-N and NO$_3^-$-N), 62.73 mg kg$^{-1}$ of available phosphorus (Olsen P), and 132.25 mg kg$^{-1}$ of available potassium.

Seeds of both tomato cultivars (D72 and Z19) were surface-sterilized using 70% ethanol for 10 s, followed by 3% sodium hypochlorite solution for 15 min, and rinsing the seeds thoroughly with sterile water. These seeds were germinated on cotton gauze at 28 °C. To prepare the split-root plants, the distal one mm of the root tip of germinated seeds was removed with a sterile scalpel, and the seeds were planted into pots (12 cm × 10 cm) filled with 700 g of sterile soil (50 KGray γ-irradiation)[60]. The absence of culturable microbes in sterilized soils was confirmed by plating soil dilutions on 50% nutrient agar Petri dishes. Twenty days later, seedlings were carefully removed from the soil and the root system was divided in two equal parts. Then, these seedlings were transplanted into the split-root set-up (Supplementary Fig. 2b), consisting of two adjacent pots (12 cm × 12 cm) filled with 800 g of sterile soil or natural soil. To prepare the unsplit-root plants, germinated seeds were planted into pots (16 cm × 14 cm) filled with 1 kg of sterile soil or natural soil. Plants were grown in a greenhouse (light/dark cycle of 16/8 h, mean day/night temperature of 30/22 °C, 70% relative humidity).

The FOL race 1 wild-type isolate FOL06 was maintained on PDA agar. The FOL conidial suspension was prepared as previously described[60,61].

### Pot experiment to test the disease resistance of tomato plants in sterile and natural soils
The pot experiment consisted of tomato plants (two genotypes) planted in four treatments: cultivar D72 grown in natural soil (*1*) or in sterile soil (*2*), cultivar Z19 grown in natural soil (*3*) or in sterile soil (*4*). Each treatment was replicated four times and each replicate consisted of 30 plants (120 plants for each treatment). The plants were maintained in the GnotoPot system as previously described[62]. In brief, the pots were placed into sterile Microboxes (0.3 m × 0.3 m × 0.6 m), and plants were irrigated with sterile water. The absence of culturable microbes in the soils was confirmed by plating soil dilutions on half-strength nutrient agar. When these plants were 60 days old, FOL conidial suspension was drenched into the soil at a final density of $1.0 \times 10^5$ conidia g$^{-1}$ soil. Thirty days later, the Fusarium disease severity was recorded and calculated using a scale containing nine grades based on the percentage of leaf yellowing/wilting[61]. In each of these treatments, the rhizosphere samples were collected to quantify FOL abundance. The rhizosphere collection was performed by gently removing the plants from the pots and removing the soil loosely attached to roots by shaking. Then, the soils tightly adhered to the roots were sampled using a sterile brush and considered as rhizosphere samples.

### Split-root system experiment to test the effects of FOL on disease severity, rhizosphere microbiota, and root exudates chemistry
Sixty-day-old split-root tomato plants grown in natural soil were used. For this experiment, two treatments for each cultivar (*i.e.*, four treatments in total, Supplementary Fig. 2a) were used: (*1*) both parts of the root system were left non-inoculated with FOL (control), and (*2*) one part of the root system was inoculated with FOL (local pot), while the other part was non-inoculated (systemic pot). FOL conidial suspension was drenched into the soil at a final density of $1.0 \times 10^5$ conidia g$^{-1}$ soil. To study the Fusarium wilt disease progression, each treatment was replicated four times and each replicate consisted of 30 plants per cultivar (120 plants for each treatment). Fusarium wilt disease symptom development was recorded at three-day intervals. To study the effects of FOL on the tomato rhizosphere microbiota and root exudates, each treatment was replicated four times and each replicate consisted of 40 plants per cultivar (160 plants for each treatment). Six days after FOL inoculation, tomato plants were harvested to collect rhizosphere samples (30 plants of each treatment in each replicate), root tissues (five plants of each treatment in each replicate), and root

exudates (five plants of each treatment in each replicate). Rhizosphere samples from the local and systemic pots were collected separately for treatments inoculated with FOL, while samples from both pots were mixed for treatments non-inoculated with FOL. One part of the rhizosphere samples was stored at −80 °C for microbiota analysis, and the other was used for the rhizosphere transplant experiment, and culturable bacteria isolation. Tomato roots were also sampled and stored at −80 °C for FOL abundance quantification. Root exudates were collected by gently removing the plants from the pots and washing the roots with sterile water. Then, the two separate parts of the root system were added into two adjacent 150-ml beakers filled with sterile deionized water containing Calcium chloride (0.5 mM). After maintaining these plants at 30°C under light for 6 h, the roots were oven-dried (60°C) and weighed, while the obtained root exudates were filtered through 0.22 μm membranes (Millipore Corp., Bedford, USA) and stored at −80°C.

### Rhizosphere transplant experiment

For the rhizosphere transplant experiment, we used the method of adding the rhizosphere inoculum to a sterile background soil[63]. The tomato rhizosphere from the split-root experiment was used as inoculum, and the sterile field soil (50 KGray γ-irradiation) as the background soil. The ratio of inoculum-to-background soil was 6% (w/w). For that, we set up eight treatments (Supplementary Fig. 2a): cultivar D72 grown in background soils mixed with rhizosphere samples of (1) non-inoculated control of D72, (2) systemic pot of FOL-inoculated D72, (3) non-inoculated control of Z19, (4) systemic pot of FOL-inoculated Z19; and cultivar Z19 grown in background soils mixed with rhizosphere samples of (5) non-inoculated control of D72, (6) systemic pot of FOL-inoculated D72, (7) non-inoculated control of Z19, (8) systemic pot of FOL-inoculated Z19. Twenty-day-old unsplit-root tomato plants prepared in sterile soil were transplanted into plastic pots (16 cm × 14 cm) containing 1 kg of these different soil mixtures (one plant per pot). Each treatment was replicated four times and each replicate consisted of 30 plants (120 plants for each treatment). Forty days after transplantation, tomato plants were inoculated with FOL as described above. Twenty-four days after FOL inoculation, the Fusarium disease severity was recorded and quantified, and rhizosphere samples were collected to quantify the FOL abundance.

### Evaluating the effects of FA on the tomato rhizosphere microbiota and root exudate chemistry

The split-root system in natural soil was used (Fig. 1g). For this experiment, a total of two treatments for each cultivar (i.e., four treatments in total) were used: (1) both parts of the root system were untreated with FA (control), and (2) one part of the root system was treated with FA (local pot), while the other part was untreated (systemic pot). Sixty-day-old tomato plants grown in natural soil were treated with FA (0.1 μg g$^{-1}$ soil). FA amendment was applied to the soil surface as a soil drench. Each treatment was replicated four times with 40 plants in each replicate (160 plants for each treatment). Six days after being treated with FA, rhizosphere samples were collected from the root system in the local pot for treatments received FA and the unamended controls (30 plants of each treatment in each replicate). One part of each rhizosphere sample was stored at −80 °C for microbiota analysis, and the other part was used for the rhizosphere transplant experiment. In addition, plant root exudates were collected (five plants of each treatment in each replicate) and analyzed using HPLC-MS (see below).

Similar as described above, a rhizosphere transplant experiment was used to test the effects of FA on the rhizosphere microbiota disease suppressiveness (Fig. 1g). For that, a total of four treatments were used: cultivar D72 grown in background soils mixed with rhizosphere samples of (1) the control of D72, (2) systemic pot of FA-treated D72; and cultivar Z19 grown in background soils mixed with rhizosphere

samples of (3) the control of Z19, (4) systemic pot of FA-treated Z19. Each treatment was replicated four times and each replicate consisted of 30 plants (120 plants for each treatment). FOL was inoculated as described above, and the Fusarium disease severity was recorded (see above). In addition, tomato rhizosphere samples were collected to quantify FOL abundance.

### Development and characterization of the FOL mutant deficient in FA production

The *FUB1* gene was deleted in the wild-type FOL as described previously[39]. Briefly, the replacement cassette containing the entire coding region of the *FUB1* gene was constructed using the fusion PCR method[64]. Primer sets *fub1*-F1/*fub1*-R1 and *fub1*-F2/*fub1*-R2 were used to amplify the upstream and downstream flanking regions of the *FUB1* gene, respectively. The primer set *fub1*-hph-F/*fub1*-hph-R was used to amplify the hygromycin B phosphotransferase cassette, under the control of the *Aspergillus nidulans gpdA* promoter and the *trpC* terminator, from the pAN7-1 plasmid. These three fragments were then PCR fused with the primers *fub1*-F1n/*fub1*-R2n. Fusion PCR products were sequenced and used for transformation. For fungal transformation, protoplasts were produced from freshly germinated conidia of FOL and transformed using the polyethylene glycol method[65]. Hygromycin B-resistant transformants were selected and further screened by PCR with primer set *fub1*-F1/*fub1*-R2 to confirm the absence of wild type *FUB1* gene. The PCR primers used are described in Supplementary Table 1. The absence of FA production in the mutant Δ*fub1* was assessed using HPLC-MS. The pathogenicity of the mutant Δ*fub1* was determined in a pot experiment. Briefly, 60-day-old unsplit-root tomato plants grown in sterile soil were inoculated with FOL or Δ*fub1* (1.0 × 10$^5$ conidia g$^{-1}$ soil), as described above. Twenty-four days after FOL inoculation, the Fusarium disease severity was recorded and calculated, as previously described (see above).

### Analyses of FOL-derived metabolites and composition of tomato root exudates

FOL-derived metabolites and tomato root exudates were analyzed using an HPLC-MS. For FOL-derived metabolites, one 5-mm agar plug of the wild-type FOL or the Δ*fub1* was taken from 7-day-old PDA agar, and then inoculated into 150 ml of potato dextrose broth. After incubation at 28 °C on a shaker at 120 rpm for ten days, the fungal biomass (mycelia and conidia) was removed by centrifugation (5,000 g) for 15 min. The supernatant (150 ml) was extracted with the same volume of 80% (v/v) acetonitrile containing 1% (v/v) acetic acid three times, then the organic phase was evaporated to dryness, dissolved in 200 μl of (v/v) 80% methanol and filtered through 0.22 μm membranes. For tomato root exudates, 200 ml of the collected solution was lyophilized, re-suspended in 2 ml of 80% (v/v) methanol, and filtered through 0.22 μm membranes. *p*-Chlorophenylalanine (1 μg ml$^{-1}$) was added as internal standard. Samples (20 μl) were analyzed by a Nexera X2 Ultra High-Performance Liquid Chromatography system (SHIMADZU, Kyoto, Japan) coupled with a 4500 QTRAP mass spectrometer (AB Sciex, Foster City, USA). For HPLC analysis, the mobile phase A (0.1% formic acid in water) and B (0.1% formic acid in acetonitrile) were used. A gradient-elution program was set as follows: 0−1 min, 95% A plus 5% B; 1−10 min, 65% A plus 35% B; 10−20 min, 50% A plus 50% B; 20−22 min, 10% A plus 90% B; 22−25 min, 90% A plus 10% B. Column temperature of 40 °C, injection volume of 4 μl and flow rate of 0.35 ml min$^{-1}$ were used. The mass spectrometer was operated as follows: ion source, turbo spray; ion temperature 550 °C; ion spray voltage 5500 V (positive mode); ion source gas I and II, 50 and 60 psi, respectively; curtain gas, 25 psi. Data were collected in multiple reaction monitoring modes. There are four biological replicates for each treatments. Eight and 32 samples were analyzed for FOL and root exudates samples, respectively. Metabolite identification was performed using exact mass and retention time from an in-house

library of metabolites (Metware Biotechnology Co., Ltd. Wuhan, China), corresponding to the second level of putative identification. Measured values for root exudates were normalized to the root dry weight.

## Quantification of FA in rhizosphere samples

The tomato rhizosphere samples (2 g) were extracted three times with 5 ml of 50% (v/v) acetonitrile containing 0.1% (v/v) formic acid, and then centrifuged at 5,000 $g$ for 10 min. The combined supernatants were evaporated to dryness, dissolved in 200 μl of (v/v) 80% methanol, and filtered through 0.22 μm membranes. Samples were analyzed using an HPLC-MS as described above. FA content was estimated based on standard curves with known concentrations of authentic compound and expressed as μg per gram of soil dry mass.

## Isolation and identification of culturable bacteria

Rhizosphere bacterial isolation was performed using a serial dilution based on an initial 1 g of sample added to 9 ml of phosphate buffer saline solution and vortexed for 2 min. Serial dilutions (i.e., $10^{-3}$ to $10^{-7}$) were plated on R2A agar, King's B agar, and 1/10 strength tryptic soy agar media. Plates were incubated for two to five days at 25 °C in the dark. Bacterial colonies were purified on tryptic soy agar. The 16 S rRNA gene of each isolate was amplified with the primer set of F27/R1492[66], and sequenced. The taxonomical identification of the isolates was determined using BLAST against the NCBI nucleotide database. In total, 39 isolates belonging to the genera *Flavobacterium*, *Arthrobacter*, *Streptomyces*, *Lysobacter*, *Sphingobium*, and *Sphingomonas* were retained in the final analysis, and a neighbor-joining phylogenetic tree was generated using IQ-TREE 2[67]. To match these isolates with OTUs from the rhizosphere microbiota sequencing data, the 16 S rRNA gene sequences of these isolates were trimmed at the sites of the primer sets F338/R806[68]. Then, the sequence similarities of each isolate with each *Flavobacterium*, *Arthrobacter*, *Streptomyces*, *Lysobacter*, *Sphingobium*, and *Sphingomonas* spp. OTUs found to be altered by FOL were determined using BLAST.

A pot experiment using 50-day-old unsplit-root tomato plants grown in sterile soil was performed to evaluate the effects of these 39 isolates on tomato Fusarium wilt disease suppression (Fig. 3b). To test the effects of each isolate, we established a total of 40 treatments for each cultivar, i.e., plants inoculated with (1) only FOL, (2-40) both FOL and each of these 39 bacterial isolates. Each treatment was replicated four times with 30 plants in each replicate (120 plants for each treatment). To test the effects of synthetic communities consisting of six selected bacterial isolates (Fl79, Ar03, St81, Ly56, Sb87, and Sm12), we established a total of 8 treatments for each cultivar, i.e., plants inoculated with (1) only FOL, (2) both FOL and the synthetic community (3–8) both FOL and the synthetic community excluding each of the six isolates independently (drop-out experiment). Each treatment was replicated four times with 30 plants in each replicate (120 plants for each treatment). The bacterial inoculations were performed as a soil drench at a final density of $1.0 \times 10^5$ CFU g$^{-1}$ soil. For each synthetic community, all isolates were mixed at a ratio of 1:1 with the total application rate kept constant. Ten days after the inoculations, FOL was inoculated as a soil drench at a final density of $1.0 \times 10^5$ conidia g$^{-1}$ soil. Fusarium wilt disease severity was measured after twenty-four days of FOL inoculation, as described above.

The in vitro antagonistic activities of the selected bacterial isolates (Fl79, Ar03, St81, Ly56, Sb87, and Sm12) against FOL were tested on PDA agar, as previously described[22].

A pot experiment using 50-day-old split-root plants grown in sterile soil was performed to test whether these isolates can induce systemic resistance in tomato plants (Fig. 3b). We established a total of seven treatments for each cultivar, i.e., one part of the root system was inoculated with FOL and the other part was either (1) untreated, or (2-7) inoculated with each of these six bacterial isolates. Bacterial isolates

were inoculated as described above ($1.0 \times 10^5$ CFU g$^{-1}$ soil). Ten days after the inoculation, FOL was inoculated ($1.0 \times 10^5$ conidia g$^{-1}$ soil). Each treatment was replicated four times with 40 plants in each replicate (160 plants for each treatment). Two days after FOL inoculation, tomato roots (the systemic roots not inoculated with the bacterial isolates) from five plants of each treatment in each replicate were sampled and stored at −80 °C for defense-related gene expression analysis (see below). Six days after FOL inoculation, rhizosphere samples were collected from five plants in each treatment per replicate to measure FOL abundance and FA content. Twenty-four days after FOL inoculation, the Fusarium disease severity of the remaining plants (30 plants in each treatment per replicate) was recorded and calculated, as previously described (see above).

## Quantification of defense-related gene expression in tomato roots

Total RNA was extracted from tomato roots using the TRIZOL reagent method (Invitrogen, Carlsbad, USA). The quality and yield of extracted RNA were checked in a 1.2% (w/v) agarose gel electrophoresis and using a NanoDrop 2000 spectrophotometer (ThermoFisher Scientific, Wilmington, USA). First-strand cDNA was synthesized with the primer oligo (dT)$_{15}$ using the TIANScript RT Kit (Tiangen Biotech, Beijing, China). The expression of *AOS* and *PR1a* genes were analyzed with SYBR Green-based quantitative reverse transcription-PCR (qRT-PCR) using the gene-specific primers described in Supplementary Table 1. The *ACTIN* gene was used as the reference 'housekeeping' gene. All amplifications were performed in triplicate. Relative expression of these genes was calculated with the $2^{-\Delta\Delta CT}$ method[69].

## Evaluating the effects of FOL, Δ*fub1*, and FA on the root colonization by bacterial isolates

Three independent experiments using the split-root system in sterile soil were performed. The first experiment tested the effects of FOL on the root colonization by the bacterial isolates (Fl79, Ar03, St81, Ly56, Sb87, and Sm12) (Fig. 4a). For that, we established a total of 14 treatments for each tomato cultivar, i.e., one part of the root system was inoculated with each of the six bacterial isolates or a synthetic community consisting of all six isolates, and the other part was either inoculated with FOL (1–7) or not (8–14). The second experiment tested the effects of FA amendment on the root colonization by the isolate Sm12 (Fig. 4a). For that, a total of six treatments for each cultivar were implemented, i.e., one part of the root system was inoculated with Sm12, and the other part was amended with FA at concentrations of 0, 0.1, 0.5, 1.0, 10, and 50 μg g$^{-1}$ soil. The third experiment tested the effects of Δ*fub1* on the root colonization by Sm12 (Fig. 4a). For that, three treatments for each cultivar were established, i.e., one part of the root system was inoculated with Sm12, and the other part was either (1) untreated or inoculated with the (2) wild-type FOL or (3) Δ*fub1*. In each of these experiments, fifty-day-old tomato plants with split-root systems grown in sterile soil were used. In brief, one part of the root system was inoculated with bacteria ($1.0 \times 10^5$ CFU g$^{-1}$ soil), and ten days later the other part was treated with FA, FOL, or Δ*fub1*. FOL and Δ*fub1* were inoculated as described above ($1.0 \times 10^5$ conidia g$^{-1}$ soil). FA amendment was applied to the soil surface as soil drench. In addition, a spontaneous rifampicin-resistant mutant of Sm12 was generated as previously described[20], and used in the second and third experiments. For each experiment, each treatment was replicated four times with nine plants in each replicate (36 plants for each treatment). Six days after FOL/Δ*fub1* inoculation or FA amendment, rhizosphere samples were collected from the root system inoculated with bacteria to quantify its abundance. The abundance of bacteria was determined using quantitative PCR in the first experiment. The cell number of Sm12 was counted by plating on LB agar medium amended with 150 μg ml$^{-1}$ rifampicin in the second and third experiments.

## Evaluating the effects of FA and Δfub1 on the tomato rhizosphere bacterial abundance in natural soil

Two independent experiments using the split-root set-up in natural soil were performed (Fig. 4a). In the first experiment, a total of six treatments for each cultivar were established, i.e., one part of the root system was untreated, and the other part was amended with FA at concentrations of 0, 0.1, 0.5, 1.0, 10, and 50 μg g$^{-1}$ soil. In the second experiment, a total of three treatments for each cultivar were established, i.e., one part of the root system was untreated, and the other part was either (1) untreated or inoculated with the (2) wild-type FOL or (3) Δfub1. Sixty-day-old tomato plants were treated with FA, FOL, or Δfub1, as described above. For each experiment, each treatment was replicated four times containing nine plants in each replicate (36 plants for each treatment). Six days after the treatment, rhizosphere samples from the root system untreated with FA, FOL, or Δfub1 were collected to quantify the bacterial abundance.

## Evaluating the potential toxicity of FA on bacterial isolates and tomato plants

To evaluate the effects of FA on the in vitro growth of the bacterial isolates (Fl79, Ar03, St81, Ly56, Sb87, and Sm12), overnight cultures of each isolate in Luria-Bertani broth were diluted to an optical density of 600 nm (OD$_{600}$) of 0.01 with fresh Luria-Bertani broth. Then, 200 μl of the diluted cell suspension was transferred into each well of a sterile 48-well polystyrene microtiter plate. After that, 20 μl of FA solution was added into each well at final concentrations of 0, 0.1, 0.5, 1.0, 10, and 50 μg ml$^{-1}$. A total of four replicates per concentration were used for each bacterial isolate. These plates were incubated at 28 °C with shaking at 125 rpm for 24 h, and the bacteria growth was determined by measuring OD$_{600}$.

The effect of FA on bacterial biofilm formation was quantified using a microtiter plate assay[70]. Briefly, overnight cultures in Luria-Bertani broth were diluted to an OD$_{600}$ of 0.01 with fresh Luria-Bertani broth. Then, 200 μl of the diluted cell suspension was transferred into each well of a sterile 48-well polystyrene microtiter plate. A volume of 20 μl of FA solution was added into each well at final concentrations of 0, 0.1, 0.5, 1.0, 10, and 50 μg ml$^{-1}$. A total of four replicates per concentration were used for each bacterial isolate. After static incubation at 28 °C for 24 h, cells adhered to the wells were stained with 1 ml of 0.1 % crystal violet for 30 min at room temperature. Biofilm formation was quantified by measuring the OD$_{600}$ of each well.

To test the potential toxicity of FA on tomato plants, sixty-day-old unsplit-root plants grown in sterile soil were treated with FA at concentrations of 0, 0.1, 0.5, 1.0, 10, and 50 μg g$^{-1}$ soil. Each concentration was repeated four times with 30 plants per replicate (120 plants for each treatment). Twenty-four days after the treatment, the wilt symptom was recorded and calculated, as previously described (see above).

## Quantification of FOL and bacterial isolate abundances

Soil genomic DNA was extracted from 0.25 g of rhizosphere samples using the Power Soil DNA Isolation Kit (MO BIO Laboratories, Carlsbad, USA). Tomato plant DNA was extracted from 0.10 g of root tissues using the Plant Genomic DNA kit (Tiangen Biotech, Beijing, China). Bacterial DNA was extracted from bacterial pellets (prepared from 2.5 ml of bacterial culture via centrifuged at 1,2000$g$ for 1 min) using the Bacteria DNA kit (Tiangen Biotech, Beijing, China). The quality and yield of extracted DNA were checked in a 1.2% (w/v) agarose gel electrophoresis and using a NanoDrop 2000 spectrophotometer (ThermoFisher Scientific, Wilmington, USA). FOL abundance was evaluated using the TaqMan quantitative PCR targeting the virulence gene SIX1 of FOL. The abundances of Flavobacterium, Arthrobacter, Streptomyces, Lysobacter, Sphingobium, and Sphingomonas spp. were determined using the SYBR Green-based quantitative PCR targeting the 16 S rRNA gene. The primers used here are listed in Supplementary Table 1.

## Amplicon sequencing and bioinformatics analysis

Briefly, the V3-V4 region of the bacterial 16S rRNA gene was amplified with the primer set F338/R806 containing specific overhang Illumina adapters[68] (Supplementary Table 1). A second eight-cycle PCR was performed to add the dual index and Illumina sequencing adapters using the Nextera XT Index Kit (Illumina Inc., San Diego, USA). The PCR products were purified, quantified, normalized, and pooled. The DNA library pool was paired-end sequenced (2 × 300) on an Illumina Miseq platform (Illumina Inc., San Diego, USA). Raw sequence reads were processed using the QIIME pipeline[71]. Chimeras were removed with USEARCH using the UCHIME algorithm[72]. Sequences were clustered into OTUs based on 97% sequence similarity using UPARSE[72]. Taxonomic affiliation of OTUs was performed using the SILVA database (v132)[73]. After removing singletons (i.e., OTUs represented by one sequence), all sequences taxonomically assigned to archaea, chloroplasts, and mitochondria were removed, and the dataset was rarefied to the minimum number of sequences per sample.

## Evaluating the effects of tomato root exudates on the growth and biofilm formation of bacterial isolates

The effects of tomato root exudates on the in vitro growth and biofilm formation of the isolate Sm12 were evaluated as similarly described for FA amendment essays (see above). To test the effects of tomato root exudates from the split-root experiment (which evaluated the effects of FOL on tomato root exudates), we established four treatments: (1) root exudates of non-inoculated controls of D72, (2) root exudates from the systemic pot inoculated with FOL in D72, (3) root exudates of non-inoculated controls of Z19, (4) root exudates from the systemic pot inoculated with FOL in Z19. To test the effects of tomato root exudates from the split-root experiment (which evaluated the effects of FA amendment on tomato root exudates), we established four treatments: (1) root exudates of untreated controls of D72, (2) root exudates from the systemic pot amended with FA in D72, (3) root exudates of untreated controls of Z19, (4) root exudates from the systemic pot amended with FA in Z19. For the evaluation of each pre-identified molecule from root exudates, each compound was tested at a final concentration of 10 μM. Each treatment was replicated four times with five wells for each replicate (20 wells for each treatment).

## Evaluating the effects of tomato root exudates on FOL growth and FA production

The wild-type FOL was grown in potato dextrose broth at 28 °C on a shaker at 125 rpm for five days. The culture was diluted to 1.0 × 10$^4$ conidia ml$^{-1}$ with fresh potato dextrose broth. Then, 500 μl of the diluted cell suspension was transferred into an Eppendorf tube containing 9 ml of fresh potato dextrose broth. After that, 500 μl of root exudates were added. The experiment consisted of four treatments: (1) root exudates of non-inoculated controls of D72, (2) root exudates from the systemic pot inoculated with FOL in D72, (3) root exudates of non-inoculated controls of Z19, (4) root exudates from the systemic pot inoculated with FOL in Z19. Each treatment was replicated four times with six tubes for each replicate (24 tubes for each treatment). The mixtures were incubated at 28 °C on a shaker at 125 rpm. After five days, the abundance of FOL in the mixture was measured by plating on potato dextrose agar, and the FA content was measured using HPLC-MS. The expression of the FUB1 gene in FOL was quantified using SYBR Green based qRT-PCR. In brief, the total RNA was extracted using the TRIZOL reagent (Invitrogen, Carlsbad, USA). The cDNA was synthesized using the FastKing RT cDNA kit (TransGen Biotech, Beijing, China). The ACTIN gene of FOL was used as the reference 'housekeeping' gene. The primers used are listed in Supplementary Table 1. All amplifications were performed in triplicate. The relative gene expression of each gene was calculated using the 2$^{-\Delta\Delta CT}$ method[69].

## Statistical analyses

All statistical analyses were performed in R (v4.2.2, http://www.r-project.org/). Normal distributions and homogeneity of variances of the data were checked using the Shapiro-Wilk test and Levene test, respectively. Significant differences in the variance of parameters were evaluated using two-sided Welch's $t$-tests (for two groups) or the one-way analysis of variance (ANOVA) followed by Tukey's HSD test (for more than two groups). Differences in tomato root exudate profiles were analyzed using the OPLS-DA analysis in the "ropls" and "mixOmics" packages. $R^2X$ and $Q^2$ indicate the goodness of fit and predictability of the OPLS-DA model, respectively. Principal component analyses (PCoA) and PERMANOVA based on Bray-Curtis dissimilarities were performed to analyze bacterial community β-diversity using the "vegan" package[74]. Differences in the relative abundance of microbial OTUs or root exudates metabolites between treatments were analyzed using the DESeq2 package[75]. $P$-values were corrected for multiple comparisons using the Benjamini-Hochberg method.

## Reporting summary

Further information on research design is available in the Nature Portfolio Reporting Summary linked to this article.

## Data availability

The amplicon sequencing data generated in this study have been deposited in the Genbank database under accession codes PRJNA882645 and PRJNA882647. The partial 16S rRNA gene sequences of the obtained bacterial isolates have been deposited in the Genbank database under accession codes PP266874-PP266912. Source data are provided with this paper.

## Code availability

The R scripts utilized for the statistical analysis and plotting of the figures are available at https://github.com/xingangzhouneau/FOL-tomato[76].

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

## Acknowledgements

This research was supported by the National Natural Science Foundation of China [42325704 (Z.W.), 32072655 (X.Z.), 32272792 (X.Z.)], the Fundamental Research Funds for the Central Universities [KYT2024001 (Z.W.)] and the USDA National Institute of Food and Agriculture and Hatch Appropriations under Project PEN04908 [7006279 (F.D.-A.)].

## Author contributions

X.Z., Z.W, J.L., and F.W. conceived and designed the experiments; X.Z., X.J., H.J., and L.R. performed the experiments and analyzed the data; X.Z., Z.W., K.S., and F.D-A. wrote the manuscript. All authors edited the manuscript and approved the final version.

## Competing interests

The authors declare no competing interests.
