## [Peer Review File · Nature Communications]

REVIEWER COMMENTS

Reviewer #1 (Remarks to the Author):

The highlight of this paper is a fusarium-acid knockout experiment. While unfortunately the author did not provide further analysis of microbiome data to verify how this regulates the entire microbiome. Overall, the paper lacks substantial innovation, and the data are limited, with only several amplicons without metagenomics, plant transcriptome and other evidences to support the conclusion.

The community data have only four replicates. It is suggested that at least 6-10 pure biological replicates should be conducted and metagenomic sequencing data also employed for further confirmation of functional gene expression.

The author used 50 KGray γ -irradiation to disinfect the soil. While it is doubtful that if a simple γ - irradiation can provide satisfactory effect. An additional TSA and PDA plating experiment is needed to convincingly demonstrate the sterilization. Could refer to Nat Protoc 6, 2450–2470 (2021).

In Figure 3a, lumping all OTUs together for species identification is not preferred as it could not reveal their identity. OTUs should first be separated by genus, and then standard 16S sequences of all type-derived seqs in each genus should be downloaded from NCBI, followed by phylogenetic tree construction separately. RAXML or IQTREE2 are also superior to Mega for phylogenetic analysis.

To verify the regulatory effect of fusaric acid on natural microflora, the supplementation of rhizosphere microbiome, meta-genomics, as well as tomato transcriptomics data of the knockout strains are needed, to analyze the community changes and to confirm the role of fusarium acid.

It is suggested to add pseudo-root experiment to further confirm the effect of fusaric acid on the recruitment of resistant and susceptible plants.

UPARSE uses 97% similarity for clustering, it is impossible to correspond the isolated bacteria to the specific OTUs. At present, most other recent studies recommend ASV or ZOTUs to better present the potential taxonomic units.

Reviewer #2 (Remarks to the Author):

The manuscript describes the results of an impressive set of complementary experiments investigating the role of fusaric acid in mediating plant response to *Fusarium* infection. These experiments suggest that susceptible and resistant genotypes of tomato display differential root exudates responses to fusaric acid released by the pathogen such that the exudates stimulate the recruitment of disease-suppressive bacterial taxa in the resistant genotype, but the opposite responses are observed in the susceptible genotype. Interestingly, the experiments were designed to highlight systemic plant responses, and these show that increased disease resistance in tomato is triggered by bacteria even when the pathogen and beneficial bacteria are spatially separated (in split-root system), suggesting that bacteria benefit plants via induced systemic responses. Overall, the study seems to be robust and well described. It provides new and in-depth information on the mechanisms of interactions between plants, pathogens and beneficial bacteria, and will be an important contribution to this research field. I have a few comments detailed below:

1. Methods description. Please clarify the numbers of replicates and total number of plants in each experiment. The methods often specify that there were four replicates per treatment and 30 plants in each replicate – what does it mean? Eg line 341: “Each treatment was replicated four times and each replicate consisted of 30 plants”. If there are four treatment combinations, does it mean there were 16 plants in total (four replicates per treatment) or $4 \times 4 \times 30 = 480$ plants? It would also be good to clearly specify the sample size in the figure legends and ideally show raw data points on the graphs in addition to means and standard errors.
2. Please write out all abbreviations in figure legends (SynComc, FA, FOL) as the figures should be understandable on their own.
3. PERMANOVA - it seems like two analyses were done – one testing the effect of genotype and the second for each genotype separately to test the effects of treatment (FOL or FA). It seems more appropriate to do a single analysis and test for the interaction between genotype and treatment – this would provide a statistical test for differences between susceptible and resistant genotypes in their responses to infection or FA.
4. OPLS-DA analysis - was this also performed in R? which package? As this is not the most used statistical tool, maybe a brief mention of what R2X and Q2 indicate might be useful.
5. The results section includes the descriptions of the experiments, which is usually not necessary to include and is repetitive of the methods. However, given that the methods are placed at the end of the manuscript, it may be useful to have these descriptions in place – please follow editorial instructions on how to format the results.
6. Tomato is a species known to form mycorrhizal associations. Is it possible that cultivars also differed in their associations in mycorrhizal fungi and could it influence their interactions with beneficial bacteria and response to pathogen attack?

7. The manuscript provides a link to github where data analysis scripts should be accessible from, but I couldn't find any scripts there.

Reviewer #3 (Remarks to the Author):

In this article entitled “Fusaric acid mediates the assembly of a disease-suppressive rhizosphere microbiota via induced shifts in plant root exudates”, the authors explored the systemic changes in plant physiology and rhizosphere microbiota of tomato plants, during infection by *Fusarium oxysporum* f. sp. *lycopersici* (FOL), associated with fusaric acid (FA) production. After challenging two tomato cultivars (D72 and Z19) with FOL in natural and sterile soil, they found that Z19 has greater resistance to the pathogen than D72 and that microbiota is a factor that mediates disease resistance. The authors used a split-root system to test the effects of FOL and FA on tomato rhizosphere microbiota in both cultivars (local pot: FOL or FA, systemic pot: non-inoculated/untreated). A transplant experiment followed, in which they found that disease severity and pathogen abundance were lower when both cultivars were grown in soil with the rhizosphere sample from previously infected resistant plants (systemic pot). Similar results were found in the experiment treated with FA. By profiling the rhizosphere bacterial community in the systemic pot, they found that FOL infection and FA amendment alter the rhizosphere microbiota in a similar way. Five isolates had the ability to reduce the symptoms of the disease in both cultivars. One of each genus was used to build a synthetic community (SynCom) and drop-out SynComs. Once they were applied in infected plants, disease severity was reduced in both cultivars, with *Sphingomonas* (Sm12) showing the greatest effect. These isolates and especially Sm12 suppressed disease severity also via induced systemic resistance. In a split-root experiment, they estimated the colonization of the isolates in response to FOL infection (one root part: isolate or SynCom, other root part: FOL inoculant). By applying FA in the local pot, with sterile or natural soil, they found that in the susceptible cultivar, Sm12 or *Sphingomonas* spp. colonization was decreased and in the resistant it was increased. They validated these results with a FOL mutant, in which no FA is produced ($\Delta fub1$). Finally, they tested in vitro effects of root exudates of FOL-inoculated or FA-treated systemic pots on growth and biofilm formation of Sm12. For the two cultivars, opposite performances were found. HPLC-MS analysis of tomato root exudates showed those metabolites that were altered and those that differentiate between the two cultivars, during FOL inoculation and FA amendment.

This research provides knowledge on the tripartite interaction of plant-pathogen-microbiota, highlighting the shifts in rhizosphere microbiota driven by the pathogen and how this affects plant health. It is a very interesting research topic, however there are some comments that need to be addressed. Please see below.

Major

1. Page 3, lines 71-73: It is true that it's more resistant when soil is natural soil, however the Z19 itself is more resistant to FOL compared to D72 irrespective of soil microbiome. Could you please give an explanation on this? Additionally, what is the proof that there is no microbiome present in sterile soil?
2. Page 3, lines 82-83: The authors mention that “FOL was only detected in the inoculated local pots but not the systemic pots (Fig. S2d)”, however, in Figure S2d this is not what is shown. Is FOL detected in systemic pots?
3. Page 4, lines 88-89: In Figure 1d, R-control soil confers high resistance in resistant plants in second generation. So, is infection indeed needed? Additionally, there are many abbreviations in the graphs, not even explained in figure legend.
4. Page 4, section “FA produced by FOL differentially affects the rhizosphere microbiota of susceptible and resistant cultivars”: It is not clear why the authors focus on the FA compound specifically. They could assess why the plants are resistant to FOL and not the contribution of FOL compound on this phenotype. That's why the authors need to better explain why FA is tested.
5. Page 4, line 97: The authors used the “split-root system and a rhizosphere transplant experiment”. However, since the soil used is natural, is the presence of other *Fusaria* expected that could contribute to FA concentrations that are be observed?
6. Page 6, line 173: The authors decide to test the potential effect of FA on root colonization by *Sphingomonas* sp. More FOL means more FA production. However, when there is more FA, SM12 is lower. Therefore, if the pathogen is there and produces FA it will suppress abundance of antagonizing microbe. Did the authors assess if application of Sm12 on the local part leads to less FA on the other side? Otherwise, the timing of this interaction is not clear. Especially if FOL infects, then Sm12 will be less.
7. Page 7, line 193: The authors mention that “ Δ fub1 mutant had weaker negative effects on the rhizosphere colonization by Sm12 and *Sphingomonas* sp.”. Nevertheless, the colonization is still high. Please comment on that.
8. Page 8, lines 219-223: I can understand that this is an interesting observation, but I miss the relevance for a publication in Nature Communications. The authors should test this in the system shown in S2a. Otherwise if a plant is infected with FOL and then recruit Sm12, does it have a chance to survive? Additionally do the different exudates affect FA biosynthetic genes in FOL in a differential manner(e.g exudates of resistant and susceptible tomato cultivar) ?
9. In general, the authors should present the story in a cleared manner. It is still puzzling how resistant and susceptible cultivars respond to FOL and then how much FA is produced by FOL during infection, and how exudation is different between them resulting in different Sm12 recruitment. These data are shown here now but their connection is weak.

Minor

1. Page 1, line 23: The authors mention “the recruitment of rhizosphere suppressive taxa”. This is like saying that these taxa suppress the rhizosphere. Maybe it is better to say “pathogen-“ or “disease-suppressive”.
2. Page 1, line 24: The authors mention “tomato root chemical exudation”. However, is the word “chemical” needed, since root exudates are chemical molecules?
3. Page 2, line 50-51 & page 3, line 61: The authors use the term “effectors” for different virulence factors. Please rephrase, as it is not clear why virulence factors are all grouped as effectors.
4. Page 3, line 59: The phrase “suppressing plants’ mechanisms associated with rhizosphere bacterial recruitment” is not very clear. Could you give an example?
5. Page 3, line 60: “plant cultivars”, please be more specific and write “tomato cultivars”.
6. Page 4, line 87: “60 days old” plants were infected with FOL. Isn’t that too old plant age for an experiment?
7. Page 5, line 135: Please specify how FOL and FA altered these OTUs. Was their abundance more or less?
8. Page 6, line 156: Please replace “systematic” with “systemic”.
9. Page 6, line 160: “tomato roots”, please specify if they are systemic roots.
10. Page 6, line 162: “Sphingomonas sp.”, if this is about the isolate, please decide whether it will be “Sphingomonas sp.” or “Sm12” in the text.
10. Page 11, line 323: “This field has been cultivated with maize for the past nine years”. How is that a relevant soil to test tomato and FOL?
11. Figure 1f: According to the text, this figure shows that “FA was detected in the rhizosphere samples of FOL-inoculated local pots but not the systemic pots”. However, the barplot shows FA content in S-control and R-control, which were not inoculated with FOL.
12. Figure 4c: Please write “FA concentration ($\mu\text{g} \cdot \text{g}^{-1}$ soil)”, instead of “Concentration ($\mu\text{g} \cdot \text{g}^{-1}$ soil)”, on the top of the barplot. Also in Figures S9a & b please write “FA concentration ($\mu\text{g} \cdot \text{ml}^{-1}$)”.
13. Tables in the Appendix (excel) are not mentioned anywhere in the text.
14. Figure S6: Please mention what the “*” indicates.

Reviewer #1 (Remarks to the Author):

The highlight of this paper is a fusarium-acid knockout experiment. While unfortunately the author did not provide further analysis of microbiome data to verify how this regulates the entire microbiome. Overall, the paper lacks substantial innovation, and the data are limited, with only several amplicons without metagenomics, plant transcriptome and other evidences to support the conclusion.

Response: We thank the reviewer for the constructive comments provided, but respectfully disagree with some arguments in this comment. In brief, we believe science quality should be based on evidence and hypothesis testing rather than the utilization of methods. With that said, our paper combines multiple experiments with clear resonance across them, all of which generated important results that unravel the dynamic modulation of disease suppressive rhizosphere microbiota across plant genotypes. Other details are addressed in our responses below.

The community data have only four replicates. It is suggested that at least 6-10 pure biological replicates should be conducted and metagenomic sequencing data also employed for further confirmation of functional gene expression.

Response: Despite the need of being reasonable when selected the proper number of biological replicates for experimentation, there is no set rule on the minimum number of replicates required for scientific research. We here used 4 replicates given the complexity of our experimental system, the lower biological variability within treatments when compared with variation across treatments, and the number of experiments performed in the manuscript. In any sense we observed the replicate number to affect our statistical power and resolution in the analyses performed. In addition to that, it is unclear how metagenomics can further confirm gene expression – as suggested by the reviewer. It seems this comment is imprecise and irrelevant to the manuscript and data we provide.

The author used 50 KGray γ -irradiation to disinfect the soil. While it is doubtful that if a simple γ -irradiation can provide satisfactory effect. An additional TSA and PDA plating experiment is needed to convincingly demonstrate the sterilization. Could refer to Nat Protoc 6, 2450–2470 (2021).

Response: Thank you for pointing this out. Indeed, we checked for the absence of culturable microbes in sterilized soils. This was performed by plating soil dilutions on 50% nutrient agar Petri dishes (as in our previous study, such as Jin et al., *Biology and Fertility of Soils*, 2020, 56:125-136). Information added in the Methods section (Lines 389-390 in the manuscript with track changes, or Lines 371-372 in the clean version).

In Figure 3a, lumping all OTUs together for species identification is not preferred as it could not reveal their identity. OTUs should first be separated by genus, and then standard 16S sequences of all type-derived seqs in each genus should be downloaded from NCBI, followed by phylogenetic tree construction separately. RAXML or IQTREE2 are also superior to Mega for phylogenetic analysis.

Response: Please, note that Figure 3a displays the phylogenetic information of the 39 bacterial isolates and did not used OTUs sequences obtained via 16S rDNA amplicon sequencing. We have now reconstructed the phylogeny using IQ-TREE 2, as suggested.

To verify the regulatory effect of fusaric acid on natural microflora, the supplementation of rhizosphere microbiome, meta-genomics, as well as tomato transcriptomics data of the knockout strains are needed, to analyze the community changes and to confirm the role of fusarium acid.

Response: We respectfully disagree with this comment. For instance, we do not see how metagenomics would allow us ‘to verify’ the data we obtained. In fact, metagenomics can be even more biased than amplicon sequencing, since only a small proportion of soil metagenomic reads (ca. 35-45% can be annotated due to database limitations). Moreover, it is unclear what “tomato transcriptomics data of the knockout strains” means. Is the reviewer suggesting that tomato transcriptomics data are necessary in our mutant FOL experiment? If so, what questions such data would answer? We believe in any moment we raised such questions, and as such, the methods utilization is not justified. In other words, we foresee how much additional work can be done as a follow up of this paper, however, we do not see the need of these additional methods, questions, experimentations, and data analysis in justifying the findings

detailed described in our study.

It is suggested to add pseudo-root experiment to further confirm the effect of fusaric acid on the recruitment of resistant and susceptible plants.

Response: Thanks for the suggestion. We further tested the effects of fusaric acid on the growth and biofilm formation of disease-suppressive taxa. We found that at high concentrations of 10 and 50 $\mu\text{g ml}^{-1}$, but not at concentrations ranging from 0.1 – 1.0 $\mu\text{g ml}^{-1}$, FA inhibited the *in vitro* growth and biofilm formation (important traits associated with rhizosphere competence and root colonization) of Sm12. Information added in Lines 213-216 in the manuscript with track changes (Lines 204-207 in the clean version).

UPARSE uses 97% similarity for clustering, it is impossible to correspond the isolated bacteria to the specific OTUs. At present, most other recent studies recommend ASV or ZOTUs to better present the potential taxonomic units.

Response: That is not always the case. Even though OTU-based approached cluster sequences at this determined level of sequence similarity (97%), sequences within a given OTU can be retrieved and aligned against other reference sequences. We believe our approach used state-of-the-art methods (since both OTUs and ASVs approaches co-exist in the literature), and that a shift to ASV would not improve in any sense the quality of our analysis or results obtained in the manuscript.

Reviewer #2 (Remarks to the Author):

The manuscript describes the results of an impressive set of complementary experiments investigating the role of fusaric acid in mediating plant response to *Fusarium* infection. These experiments suggest that susceptible and resistant genotypes of tomato display differential root exudates responses to fusaric acid released by the pathogen such that the exudates stimulate the recruitment of disease-suppressive bacterial taxa in the resistant genotype, but the opposite responses are observed in the susceptible genotype. Interestingly, the experiments were designed to highlight systemic plant responses, and these show that increased disease resistance in tomato is triggered by bacteria even when the pathogen and beneficial bacteria are spatially separated (in split-root system), suggesting that bacteria benefit plants via induced systemic responses. Overall, the study seems to be robust and well described. It provides new and in-depth information on the mechanisms of interactions between plants, pathogens and beneficial bacteria, and will be an important contribution to this research field. I have a few comments detailed below:

Response: Thanks for the positive assessment of our manuscript.

1. Methods description. Please clarify the numbers of replicates and total number of plants in each experiment. The methods often specify that there were four replicates per treatment and 30 plants in each replicate – what does it mean? e.g. line 341: “Each treatment was replicated four times and each replicate consisted of 30 plants”. If there are four treatment combinations, does it mean there were 16 plants in total (four replicates per treatment) or $4 \times 4 \times 30 = 480$ plants? It would also be good to clearly specify the sample size in the figure legends and ideally show raw data points on the graphs in addition to means and standard errors.

Response: Thank you for pointing this out. Information added in the main manuscript. e.g. “Each treatment was replicated four times and each replicate consisted of 30 plants”. As such, each treatment consisted of 120 plants”. We have also included the sample size in the figure legends and show raw data points on the graphs.

2. Please write out all abbreviations in figure legends (SynComc, FA, FOL) as the figures should be understandable on their own.

Response: Corrected accordingly.

3. PERMANOVA - it seems like two analyses were done – one testing the effect of genotype and the second for each genotype separately to test the effects of treatment (FOL or FA). It seems more appropriate to do a

single analysis and test for the interaction between genotype and treatment – this would provide a statistical test for differences between susceptible and resistant genotypes in their responses to infection or FA.

Response: We have now performed a two-way PERMANOVA to test the effects of genotype and treatment (FOL or FA), in addition to their interaction on bacterial community beta-diversity. The manuscript was revised accordingly (Lines 121-124, 143-146 in the manuscript with track changes, or Lines 118-120, 138-140 in the clean version).

4. OPLS-DA analysis - was this also performed in R? which package? As this is not the most used statistical tool, maybe a brief mention of what R2X and Q2 indicate might be useful.

Response: The OPLS-DA analysis was performed using the R packages “ropls” and “mixOmics”. The R^2X and Q^2 parameters indicate the goodness of fit and the predictability of the OPLS-DA model, respectively. The manuscript was revised accordingly (Lines 671-673 in the manuscript with track changes, or Lines 643-645 in the clean version).

5. The results section includes the descriptions of the experiments, which is usually not necessary to include and is repetitive of the methods. However, given that the methods are placed at the end of the manuscript, it may be useful to have these descriptions in place – please follow editorial instructions on how to format the results.

Response: Thank you for the note. Indeed, having a short description of the methods is indicated to better guide the readability of the text. We revised the text accordingly to avoid repetition and details of the methods in the Results section.

6. Tomato is a species known to form mycorrhizal associations. Is it possible that cultivars also differed in their associations in mycorrhizal fungi and could it influence their interactions with beneficial bacteria and response to pathogen attack?

Response: Thank you for highlighting this important factor in the system. Despite we did not test for the potential effect of AMF associations in our system, we have now included a sentence in the Discussion to address this point. Please, see Lines 303-306 in the manuscript with track changes (or Lines 287-290 in the clean version): It is important to notice that our study specifically focused on rhizosphere bacterial communities. As such, other types of biotic interactions in the rhizosphere, including fungal communities and mycorrhizal associations – known to differ across plant genotypes and potentially affect disease outcomes^{49,50} – were not explicitly studied in our system.

7. The manuscript provides a link to github where data analysis scripts should be accessible from, but I couldn't find any scripts there.

Response: Correction made. The link is now active for full access to our data and scripts.

With best regards,

Marina Semchenko

Reviewer #3 (Remarks to the Author):

In this article entitled “Fusaric acid mediates the assembly of a disease-suppressive rhizosphere microbiota via induced shifts in plant root exudates”, the authors explored the systemic changes in plant physiology and rhizosphere microbiota of tomato plants, during infection by *Fusarium oxysporum* f. sp. *lycopersici* (FOL), associated with fusaric acid (FA) production. After challenging two tomato cultivars (D72 and Z19) with FOL in natural and sterile soil, they found that Z19 has greater resistance to the pathogen than D72 and that microbiota is a factor that mediates disease resistance. The authors used a split-root system to test the effects of FOL and FA on tomato rhizosphere microbiota in both cultivars (local pot: FOL or FA, systemic pot: non-inoculated/untreated). A transplant experiment followed, in which they found that disease severity and pathogen abundance were lower when both cultivars were grown in soil with the rhizosphere sample from previously infected resistant plants (systemic pot). Similar results were found in the experiment treated with

FA. By profiling the rhizosphere bacterial community in the systemic pot, they found that FOL infection and FA amendment alter the rhizosphere microbiota in a similar way. Five isolates had the ability to reduce the symptoms of the disease in both cultivars. One of each genus was used to build a synthetic community (SynCom) and drop-out SynComs. Once they were applied in infected plants, disease severity was reduced in both cultivars, with *Sphingomonas* (Sm12) showing the greatest effect. These isolates and especially Sm12 suppressed disease severity also via induced systemic resistance. In a split-root experiment, they estimated the colonization of the isolates in response to FOL infection (one root part: isolate or SynCom, other root part: FOL inoculant). By applying FA in the local pot, with sterile or natural soil, they found that in the susceptible cultivar, Sm12 or *Sphingomonas* spp. colonization was decreased and in the resistant it was increased. They validated these results with a FOL mutant, in which no FA is produced (Δ fub1). Finally, they tested in vitro effects of root exudates of FOL-inoculated or FA-treated systemic pots on growth and biofilm formation of Sm12. For the two cultivars, opposite performances were found. HPLC-MS analysis of tomato root exudates showed those metabolites that were altered and those that differentiate between the two cultivars, during FOL inoculation and FA amendment.

This research provides knowledge on the tripartite interaction of plant-pathogen-microbiota, highlighting the shifts in rhizosphere microbiota driven by the pathogen and how this affects plant health. It is a very interesting research topic, however there are some comments that need to be addressed. Please see below.

Response: Thank you for the positive assessment of our manuscript. Below we provide point-by-point responses to the comments provided.

Major

1. Page 3, lines 71-73: It is true that it's more resistant when soil is natural soil, however the Z19 itself is more resistant to FOL compared to D72 irrespective of soil microbiome. Could you please give an explanation on this? Additionally, what is the proof that there is no microbiome present in sterile soil?

Response: Thank for pointing this out. Indeed, we found the resistant cultivar to have a higher resistant to FOL when compared to the susceptible cultivar irrespective of soil microbiota. We believe this might be due to variation in plant innate immunity that contributes to FOL resistance across cultivars. We have now added this argument in the Discussion section (Lines 293-295 in the manuscript with track changes, or Lines 277-279 in the clean version).

Regarding the soil sterility, this was checked by plating sterile soil serial dilutions in on 50% nutrient agar medium. No culturable bacteria was detected. Information added in the Methods section (Lines 389-390 in the manuscript with track changes, or Lines 371-372 in the clean version).

2. Page 3, lines 82-83: The authors mention that “FOL was only detected in the inoculated local pots but not the systemic pots (Fig. S2d)”, however, in Figure S2d this is not what is shown. Is FOL detected in systemic pots?

Response: Revised accordingly. In brief, FOL was not detected in systemic pots. Check Fig. S2d.

3. Page 4, lines 88-89: In Figure 1d, R-control soil confers high resistance in resistant plants in second generation. So, is infection indeed needed? Additionally, there are many abbreviations in the graphs, not even explained in figure legend.

Response: One of the main aims of our study was to test the effect of FOL and its virulence factor (fusaric acid) on the tomato rhizosphere microbiota. Our results revealed that FOL infection and fusaric acid treatment increased the disease suppressiveness (via rhizosphere microbiota modulation) in the resistant cultivar but decreased it in the susceptible cultivar. The rhizosphere microbiota from the resistant cultivar not infected with FOL had higher disease suppressiveness than that of the susceptible cultivar not infected with FOL, which were consistent with previous reports (Line 54-56 in the manuscript with track changes, or Lines 53-55 in the clean version). Our new finding is that the FOL-induced changes in the rhizosphere microbiota differed between the resistant and susceptible tomato cultivars. Thus, the infection is needed.

All abbreviations are now detailed in the figure legends.

4. Page 4, section “FA produced by FOL differentially affects the rhizosphere microbiota of susceptible and resistant cultivars”: It is not clear why the authors focus on the FA compound specifically. They could assess why the plants are resistant to FOL and not the contribution of FOL compound on this phenotype. That's why the authors need to better explain why FA is tested.

Response: Thank you for the note. In fact, fusaric acid is a secondary metabolite secreted by several pathogenic *Fusarium* species. This compound is known to exhibit phytotoxic activity and induces wilt symptoms in plants. Therefore, fusaric acid constitutes a well-known virulence factor. Emerging evidence indicates that plant pathogens can often utilize virulence factors to alter the host rhizosphere microbiota and prime the disease development. Together, these are lines of justification for our choice in studying the effects of FA on tomato rhizosphere microbiota. We have now included these arguments in the Introduction section (Lines 61-64 in the manuscript with track changes, or Lines 60-63 in the clean version).

5. Page 4, line 97: The authors used the “split-root system and a rhizosphere transplant experiment”. However, since the soil used is natural, is the presence of other *Fusaria* expected that could contribute to FA concentrations that are observed?

Response: Great point! In fact, in the split-root experiment used to test the effects of FOL on tomato performance, we did not detect FA in the control treatment and in the local pots. Thus, the potential effects of the presence of other *Fusaria* are either absent or minor. Information added to the main text (Lines 110-111 in the manuscript with track changes, or Lines 107-108 in the clean version).

6. Page 6, line 173: The authors decide to test the potential effect of FA on root colonization by *Sphingomonas* sp. More FOL means more FA production. However, when there is more FA, Sm12 is lower. Therefore, if the pathogen is there and produces FA it will suppress abundance of antagonizing microbe. Did the authors assess if application of Sm12 on the local part leads to less FA on the other side? Otherwise, the timing of this interaction is not clear. Especially if FOL infects, then Sm12 will be less.

Response: This is a great point. In brief, we measured the FA concentration in the tomato rhizosphere from the split-root experiment used to test the effects of bacterial isolates inoculation on wilt disease resistance. The results show that the inoculation with disease-suppressive bacterial taxa on one side of the root system resulted in lower FA concentration on the other side. Information added in the revised manuscript (Lines 185-186 in the manuscript with track changes, or Lines 177-179 in the clean version).

7. Page 7, line 193: The authors mention that “ $\Delta fub1$ mutant had weaker negative effects on the rhizosphere colonization by Sm12 and *Sphingomonas* sp.”. Nevertheless, the colonization is still high. Please comment on that.

Response: Great point! Compared to the non-inoculated control, the $\Delta fub1$ resulted in lower abundances of Sm12 and *Sphingomonas* sp. in the susceptible cultivar, and in greater abundances in the resistant cultivar. However, compared with the FOL wild type, the $\Delta fub1$ mutant had weaker negative effects on the rhizosphere colonization by Sm12 and *Sphingomonas* sp. in the susceptible cultivar and weaker positive effects on the colonization in the resistant cultivar. The main comparison here is between the wild-type and the $\Delta fub1$. Our results indicate that other virulence factors of FOL also modulate the root colonization by plant-beneficial bacteria. These points were carefully revised in the manuscript (Lines 334-337 in the manuscript with track changes, or Lines 317-320 in the clean version).

8. Page 8, lines 219-223: I can understand that this is an interesting observation, but I miss the relevance for a publication in Nature Communications. The authors should test this in the system shown in S2a. Otherwise if a plant is infected with FOL and then recruit Sm12, does it have a chance to survive? Additionally do the different exudates affect FA biosynthetic genes in FOL in a differential manner (e.g. exudates of resistant and susceptible tomato cultivar)?

Response: Specifically for this experiment, root exudates were sampled from the split-root system (information now added to the manuscript). In addition, we also tested the effects of root exudates on the FOL growth and FA production. In brief, the results showed that root exudates from the resistant and susceptible cultivars had similar effects on FOL growth, FA production and the expression of the *FUB1* gene. Information revised in the main text (Lines 254-256 in the manuscript with track changes, or Lines

245-247 in the clean version).

9. In general, the authors should present the story in a cleared manner. It is still puzzling how resistant and susceptible cultivars respond to FOL and then how much FA is produced by FOL during infection, and how exudation is different between them resulting in different Sm12 recruitment. These data are shown here now but their connection is weak.

Response: Thanks for this comment. We have now revised the entire text for clarity and readability. Here, we show that a key molecule (FA) produced by the pathogen FOL can result in distinct impacts on the modulation of the tomato rhizosphere microbiota in susceptible and resistant cultivars. In brief, we found that – at the early stages of infection – FOL-derived FA was higher in the rhizosphere of susceptible cultivar than in that of the resistant cultivar. We further explored this to show that FA content differentially modulates the composition of root exudates across cultivars and that such effects directly affect the plant’s ability to recruit beneficial microbial taxa in the rhizosphere. As such, we here report an indirect effect of FOL-derived FA in structuring a rhizosphere disease-suppressive or conducive microbiota (Fig. 5e), with trackable variations across susceptible and resistant cultivars. Information revised in the main text (Lines 261-268 in the manuscript with track changes, or Lines 252-259 in the clean version).

Minor

1. Page 1, line 23: The authors mention “the recruitment of rhizosphere suppressive taxa”. This is like saying that these taxa suppress the rhizosphere. Maybe it is better to say “pathogen-“ or “disease-suppressive”.

Response: We have now replaced with “the recruitment of rhizosphere disease-suppressive taxa”.

2. Page 1, line 24: The authors mention “tomato root chemical exudation”. However, is the word “chemical” needed, since root exudates are chemical molecules?

Response: The word “chemical” was removed as suggested.

3. Page 2, line 50-51 & page 3, line 61: The authors use the term “effectors” for different virulence factors. Please rephrase, as it is not clear why virulence factors are all grouped as effectors.

Response: We removed the term “effectors” here, and now used the term “virulence factors” throughout the manuscript.

4. Page 3, line 59: The phrase “suppressing plants’ mechanisms associated with rhizosphere bacterial recruitment” is not very clear. Could you give an example?

Response: Revised accordingly, now it reads “suppressing plants’ ability to recruit plant-beneficial microbes”.

5. Page 3, line 60: “plant cultivars”, please be more specific and write “tomato cultivars”.

Response: Revised accordingly.

6. Page 4, line 87: “60 days old” plants were infected with FOL. Isn’t that too old plant age for an experiment?

Response: The date count was based on the day seeds germinated. In this case, at 60 days, tomato plants had six leaflets, which was ideal for our experimental system.

7. Page 5, line 135: Please specify how FOL and FA altered these OTUs. Was their abundance more or less?

Response: FOL and FA resulted in the detection of lower relative abundances of the exact same OTUs in the susceptible cultivar, all of which were detected at higher relative abundances in the resistant cultivar. These OTUs were taxonomically assigned to *Lysobacter*, *Sphingobium*, and *Sphingomonas* spp.

This information was revised in the main text (Lines 155-158 in the manuscript with track changes, or Lines 148-151 in the clean version).

8. Page 6, line 156: Please replace “systematic” with “systemic”.

Response: Correction made.

9. Page 6, line 160: “tomato roots”, please specify if they are systemic roots.

Response: Yes, these were systemic roots not inoculated with the bacterial isolates. Corrected accordingly.

10. Page 6, line 162: “Sphingomonas sp.”, if this is about the isolate, please decide whether it will be “Sphingomonas sp.” or “Sm12” in the text.

Response: We now use “Sm12” here and throughout the text when mentioning the isolate.

10. Page 11, line 323: “This field has been cultivated with maize for the past nine years”. How is that a relevant soil to test tomato and FOL?

Response: The point here was to use a soil not previously cultivated with tomato plants to prevent potential effects of high FOL densities in our experimental systems (Lines 379-380 in the manuscript with track changes, or Lines 362-363 in the clean version).

11. Figure 1f: According to the text, this figure shows that “FA was detected in the rhizosphere samples of FOL-inoculated local pots but not the systemic pots”. However, the barplot shows FA content in S-control and R-control, which were not inoculated with FOL.

Response: These sample names were corrected, and now it shows S-Local and R-Local.

12. Figure 4c: Please write “FA concentration ($\mu\text{g} \cdot \text{g}^{-1}$ soil)”, instead of “Concentration ($\mu\text{g} \cdot \text{g}^{-1}$ soil)”, on the top of the barplot. Also in Figures S9a & b please write “FA concentration ($\mu\text{g} \cdot \text{ml}^{-1}$)”.

Response: Revised accordingly.

13. Tables in the Appendix (excel) are not mentioned anywhere in the text.

Response: Thank you for pointing this out. These tables were removed.

14. Figure S6: Please mention what the “*” indicates.

Response: “*” represents statistically significant differences (Wald test, Benjamini-Hochberg adjusted $P < 0.01$). Information added in the figure legend.

REVIEWERS' COMMENTS

Reviewer #1 (Remarks to the Author):

Through a fine-designed experiment, this paper highlights the importance of the tripartite interaction

of plant-fungal pathogen-microbiota in maintaining plant health. The results of the study revealed that FOL infection led to changes in the rhizosphere microbiota and that the influence on the rhizosphere microbiota differed between susceptible and resistant cultivars. Resistant cultivars recruited disease-suppressive rhizosphere microbiota after FOL infection, while the, FOL weakened the ability of susceptible plants to recruit disease-resistant microbes. These FOL-induced changes in the tomato rhizosphere microbiota were largely mediated by fusaric acid (FA), a secondary metabolite secreted by *Fusarium* species, which altered the root exudates of the plants (but differentially between cultivars). Overall the paper address interesting topic, with generally robust supporting data, which will be an important reference to the field of plant-pathogen-microbiota interaction. The several technical issues raised in the review have been satisfactorily revised, or well rationally justified by the author. I think the paper is in a good shape to be accepted for publication.

Reviewer #2 (Remarks to the Author):

I have no further comments.

Reviewer #3 (Remarks to the Author):

We thank the authors for carefully revising their manuscript and addressing previous comments and suggestions. However, there are minor comments to be addressed. Please see below.

Lines 246-247: “In addition, FA amendment did not result in any effects on tomato root exudates with implications for FOL growth and FA production (Supplementary Fig. 11b)”. This does not read well. The effect is from the tomato root exudates (which evaluated the effects of FA amendment on tomato root exudates) on FOL growth, FA production and the expression of FUB1 gene, which was similar for plants treated with FA or not. Please rephrase.

Lines 278-279: “We believe this might be due to variation in plant innate immunity that contributes to FOL resistance across cultivars”. Example/citation is needed.

Reviewer #1 (Remarks to the Author):

Through a fine-designed experiment, this paper highlights the importance of the tripartite interaction of plant-fungal pathogen-microbiota in maintaining plant health. The results of the study revealed that FOL infection led to changes in the rhizosphere microbiota and that the influence on the rhizosphere microbiota differed between susceptible and resistant cultivars. Resistant cultivars recruited disease-suppressive rhizosphere microbiota after FOL infection, while the, FOL weakened the ability of susceptible plants to recruit disease-resistant microbes. These FOL-induced changes in the tomato rhizosphere microbiota were largely mediated by fusaric acid (FA), a secondary metabolite secreted by *Fusarium* species, which altered the root exudates of the plants (but differentially between cultivars). Overall the paper address interesting topic, with generally robust supporting data, which will be an important reference to the field of plant-pathogen-microbiota interaction. The several technical issues raised in the review have been satisfactorily revised, or well rationally justified by the author. I think the paper is in a good shape to be accepted for publication.

Response: Thanks for the positive assessment of our manuscript.

Reviewer #2 (Remarks to the Author):

I have no further comments.

Response: Thanks for the positive assessment of our manuscript.

Reviewer #3 (Remarks to the Author):

We thank the authors for carefully revising their manuscript and addressing previous comments and suggestions. However, there are minor comments to be addressed. Please see below.

Lines 246-247: “In addition, FA amendment did not result in any effects on tomato root exudates with implications for FOL growth and FA production (Supplementary Fig. 11b)”. This does not read well. The effect is from the tomato root exudates (which evaluated the effects of FA amendment on tomato root exudates) on FOL growth, FA production and the expression of *FUB1* gene, which was similar for plants treated with FA or not. Please rephrase.

Response: We thank the reviewer for the constructive comments provided. This sentence is revised as “In addition, FA amendment did result in changes in tomato root exudates with potential subsequent effects on FOL growth, FA production, and *FUB1* gene expression (Supplementary Fig. 11b).”

Lines 278-279: “We believe this might be due to variation in plant innate immunity that contributes to FOL resistance across cultivars”. Example/citation is needed.

Response: Reference added as requested:

Wang, Y., Pruitt R. N., Nürnberger T. & Wang Y. Evasion of plant immunity by microbial pathogens. *Nat. Rev. Microbiol.* 20, 449-464 (2022).

Ma, L. J. et al. *Fusarium* pathogenomics. *Annu. Rev. Microbiol.* 67, 399-416 (2013).